

# An evapotranspiration model self-calibrated from remotely sensed surface soil moisture, land surface temperature and vegetation cover fraction: application to disaggregated SMOS and MODIS data

Bouchra Ait Hssaine [1,2], Olivier Merlin [2], Jamal Ezzahar [3], Nitu Ojha [2], Salah Er-raki [4], and Said Khabba [1]

[1]LMME,Département de physique, Faculté des sciences Semlalia, Université Cadi Ayyad, Marrakech, Morocco
[2]CESBIO, Université de Toulouse, IRD/CNRS/UPS/CNES, Toulouse, France
[3]Département d'Informatique et Télécommunications, MTI, Ecole Nationale des Sciences Appliquées, Université Cadi Ayyad, Safi, Morocco
[4]LP2M2E, Département de Physique Appliquée, Faculté des Sciences et Techniques, Université Cadi Ayyad, Marrakech, Morocco

Correspondence: Bouchra Ait Hssaine
(bouchraaithssaine@gmail.com)

**Abstract.** Thermal-based two-source energy balance modeling is very useful for estimating the land evapotranspiration (ET) at a wide range of spatial and temporal scales. However, the land surface temperature ($LST$) is not sufficient for constraining simultaneously both soil and vegetation flux components in such a way that assumptions (on either the soil or the vegetation fluxes) are commonly required. To avoid such assumptions, a new energy balance model TSEB-SM was recently developed

5    in (Ait Hssaine et al., 2018b) to integrate the microwave-derived near-surface soil moisture (SM), in addition to the thermal-derived $LST$ and fractional vegetation cover ($f_c$). Whereas, TSEB-SM has been recently tested using in-situ measurements, the objective of this paper is to evaluate the performance of TSEB-SM in real-life using 1 km resolution MODIS (Moderate resolution imaging spectroradiometer) $LST$ and $f_c$ data and the 1 km resolution $SM$ data disaggregated from SMOS (Soil Moisture and Ocean Salinity) observations by using DisPATCh. The approach is applied during a four-year period (2014-

10    2018) over a rainfed wheat field in the Tensift basin, central Morocco, during a four-year period (2014-2018). The field was seeded for the 2014-2015 (S1), 2016-2017 (S2) and 2017-2018 (S3) agricultural season, while it was not ploughed(remained as bare soil) during the 2015-2016 (B1) agricultural season. The mean retrieved values of ($a_{rss}$,$b_{rss}$) calculated for the entire study period using satellite data are (7.32, 4.58). The daily calibrated $\alpha_{PT}$ ranges between 0 and 1.38 for both S1 and S2. Its temporal variability is mainly attributed to the rainfall distribution along the agricultural season. For S3, the daily retrieved

15    $\alpha_{PT}$ remains at a mostly constant value ($\sim$0.7) throughout the study period, because of the lack of clear sky disaggregated $SM$ and $LST$ observations during this season. Compared to eddy covariance measurements, TSEB driven only by $LST$ and $f_c$ data significantly overestimates latent heat fluxes for the four seasons. The overall mean bias values are 119, 94, 128 and 181 W/$m^2$ for S1, S2, S3 and B1 respectively. In contrast, these errors are much reduced when using TSEB-SM (SM and $LST$ combined data) with the mean bias values estimated as 39, 4, 7 and 62 W/$m^2$ for S1, S2, S3 and B1 respectively.





# 1   Introduction

Evapotranspiration (ET) is a crucial water flux in semi-arid areas where it strongly impacts the drought monitoring, water resource management and climate (Littell et al., 2016; Molden et al., 2010). Furthermore, a precise estimate of ET determines the crop water requirements, which subsequently allows the irrigation regimes to be optimized (Allen et al., 1998).

The most commonly used method to estimate ET at the field scale is the FAO-56 method (Allen et al., 2000, 1998) forced by a vegetation index (Bausch and Neale, 1990). FAO-56 is based on the concepts of reference evapotranspiration (ET0) and crop coefficients ($k_c$). A limitation of this method is that the coefficients used (in the form of a single or double coefficient) are generally calibrated from in-situ measurements. Therefore, given the natural variability of soil properties and vegetation types, as well as hydro-meteorological and soil moisture conditions at the landscape scale (Choi et al., 2009), it is difficult to scale up

ET at regional scales from the FAO-56 method alone.

Regarding the data availability over extended areas, remote sensing is the only viable technique that can provide representative and multi-resolution measurements of ET. As a consequence, the spatial modelling has become a dominant means to estimate ET fluxes over regional and continental areas (Anderson et al., 2007; Fisher et al., 2017). One of the most widely used ET spatial models is the temperature-based approach. Models based on land surface temperature (LST) data classified into two

main groups: (i) residual balance methods and (ii) contextual methods. The first category considers ET as the residual term of the energy balance. In particular, one can cite two commonly used models: TSEB (Two-Source Energy Balance, (Norman et al., 1995)) and SEBS (Surface Energy Balance System, (Su, 2002)). The second category of models estimates ET as the potential ET times the evaporative efficiency (Moran et al., 1994) or as the available energy times the evaporative fraction (Merlin et al., 2013; Roerink et al., 2000). These models are called "contextual" because the evaporative fraction involves determining (or

extrapolating) the extreme values of $LST$ for dry and wet conditions.

Among well-known temperature-driven energy flux models, the TSEB model proposed by Norman et al. (1995) has been shown to be robust for a wide range of landscapes (Colaizzi et al., 2012; Ait Hssaine et al., 2018a). TSEB has two key input variables, which can be derived from remote sensing data. The first one is the $LST$ and the second is vegetation cover fraction ($f_c$). The TSEB model adopts an iterative procedure, in which an initial estimate of the plant transpiration is given by the

Priestly-Taylor (PT) formulation (Priestley and Taylor, 1972). This assumption requires few input data and allows a precise estimate of potential ET (Fisher et al., 2008). Nevertheless, several studies (Ait Hssaine et al., 2018b; Fisher et al., 2008; Jin et al., 2011; Yang et al., 2015) have stressed that the PT coefficient cannot be considered as a constant value, as it is influenced by several parameters. Other authors (Gonzalez-dugo et al., 2009; Long and Singh, 2012; Morillas et al., 2014) reported that the PT approach may overestimate the canopy ET, especially for low soil wetness, and/or sparse vegetation

cover, because it does not include a reasonable reduction of the initial canopy ET under stress conditions. Recently, Boulet et al. (2015) have developed the Soil-Plant-Atmosphere and Remote Sensing Evapotranspiration (SPARSE) model similar to TSEB model in its basic assumption. Nevertheless, SPARSE is solved in two modes: the prescribed and the retrieval mode to constrain the output fluxes. The former first generates an equilibrium $LST$ from the evaporation efficiency and the transpiration efficiency estimates by assuming that their values are equal to 1. Then, $LST$ is implemented in the SPARSE retrieval mode to





circumscribe the output fluxes by both limiting cases (namely the fully stressed and potential conditions). In spite of the good retrieval performances of ET by this model significant uncertainties are observed during the quasi-senescent vegetation period (Boulet et al., 2015) . Alternatively to the use of $LST$ as a proxy for ET, numerous studies have stressed that the soil moisture plays a critical role in the partitioning of available energy into latent and sensible heat fluxes and is the prominent controlling

factor of actual ET (Boulet et al., 2015; Gokmen et al., 2012; Kustas et al., 1998, 1999; Li et al., 2006). Several authors have revised the well-known LST-based TSEB model and replaced the $LST$ with microwave-derived surface soil moisture (SM) to estimate daily ET (Bindlish et al., 2001; Kustas et al., 1998, 1999; Li et al., 2006). Bindlish et al. (2001) found that the impact of $SM$ on surface fluxes is strongly related to the vegetation cover. The impact is high for low fraction cover, and relatively weak for high cover fraction. Moreover, the soil evaporation is constrained by the $SM$ through soil-texture dependent coefficients

reported in (Sellers et al., 1992). In the same way, Li et al. (2006) indicated that the model performance is sensitive to these two coefficients, and thus they proposed to average the output of LST-based TSEB and SM-based TSEB models, in order to provide more consistent results over a wide range of conditions. More recently, Yao et al. (2017) evaluated three satellite-based PT algorithms (ATI-PT, VPD-PT and SM-PT for apparent thermal inertia-, vapour pressure deficit- and SM-based formulations of the PT coefficients, respectively) to estimate terrestrial water flux in different biomes. Their finding showed that the SM-PT

algorithm had relatively better results compared to those of ATI-PT and VPD-PT. However, all three models underestimated ET in irrigated crops, reflecting that those algorithms may not capture well the soil evaporation, notably through its (site-specific) parameterization with SM. In the same vein, (Purdy et al., 2018) updated the PT Jet Propulsion Laboratory (PT-JPL) and incorporated the $SM$ data derived from SMAP (Soil Moisture Active and Passive, (Entekhabi et al., 2010)) to constrain both evaporation and transpiration, separately. The model showed high improvements compared to the original PT-JPL, especially

in dry conditions. However, the model relied on evaporation and transpiration reduction parameters, whose values were set a priori.

Previous studies, either $LST$- or SM-based, agree with the view that combining both $LST$ and $SM$ information at a time would enhance the robustness and accuracy of ET estimates in various biomes and climates. Nevertheless, few studies have simultaneously combined both observations in a unique energy balance model. One difficulty lies in developing a consistent

representation of the soil evaporation (as constrained by SM, (Chanzy and Bruckler, 1993)), the total ET (as constrained by $LST$, (Norman et al., 1995)) and the plant transpiration (as indirectly constrained by both $LST$ and SM, (Ait Hssaine et al., 2018b)). Gokmen et al. (2012) explicitly integrated the $SM$ derived from AMSR-E (Advanced Microwave Scanning Radiometer for EOS) data (Owe et al., 2008) into the LST-derived SEBS model via the kB-1 parameter, which plays an important role in the aerodynamic resistance. The updated SEBS model (SEBS-SM) provided a large improvement of sensible

heat flux and thus ET estimates under water-limited conditions. The point is that the soil evaporation reduction parameters were calibrated using in-situ measurements, which limits the validity of the approach over large areas. In the same vein, Gan and Gao (2015) incorporated a SM-based soil resistance term in the TSEB formalism and calibrated several parameters (including the PT coefficient) using $LST$ data. The obtained results showed that the model calibrated by $LST$ data performed better than the non-calibrated one. Note that the parameters of the soil resistance were set to constant values as in Sellers et al. (1992). As

a further step towards the combination of $LST$ and $SM$ data. Ait Hssaine et al. (2018b) modified the TSEB formalism (Kustas





and Norman, 1999; Norman et al., 1995) (named TSEB-SM) and proposed a new calibration strategy of the main PT-based TSEB-SM parameters. The TSEB-SM model was tested using in-situ measurements and provided an important improvement in terms of latent heat flux/sensible heat flux estimates compared to the classic TSEB all along the agricultural season, especially during the crop emergence and the senescence periods. Such improvements are attributed to stronger constraints exerted on the representation of soil evaporation (via $SM$ data and the calibrated soil parameters) and plant transpiration (via the calibrated daily PT coefficient). One crucial point is that all the above studies based on remotely sensed $SM$ and $LST$ have neglected the mismatch in the spatial resolutions of readily available $SM$ products. Especially, the global scale $SM$ data sets have a typical resolution of 40-50 km (Entekhabi et al., 2010; Kerr et al., 2010; Njoku et al., 2003). Such spatial resolution is generally unsuitable or even incompatible with many hydrological and agricultural applications. To fill the gap, disaggregation approaches of AMSR-E, SMOS and SMAP like $SM$ data have been developed Peng et al. (2017) but, to date, there has been no application of SM-based ET models to disaggregate $SM$ data sets. In addition, the use of remote sensing data would be necessary in order to avoid the time-consuming process of calibrating the TSEB model over each field.

The objective of this study is to investigate how satellite data can be used to retrieve the main parameters ($a_{rss}$, $b_{rss}$ and $\alpha_{PT}$) of TSEB-SM model. In this purpose, TSEB-SM is applied to 1 km resolution using MODIS (Moderate resolution imaging spectroradiometer) LST/ $f_c$ data and to SMOS $SM$ data is applied. To make the SMOS data spatially consistent with MODIS data, the SMOS $SM$ is disaggregated at 1 km resolution using the DisPATCh (DISaggregation based on Physical And Theoretical scale Change) algorithm (Malbéteau et al., 2016; Merlin et al., 2013; Molero et al., 2016). The proposed methodology is evaluated over a rainfed wheat field in the Tensift basin, central Morocco during four agricultural seasons (2014-2018).

## 2 Data description and methods

### 2.1 Site and in-situ data description

The study site is situated in the east (Sidi Rahal) of the Tensift basin in central Morocco (see Figure 1). The region is characterized by a semi-arid Mediterranean climate, with an average yearly precipitation of about 250 mm and an atmospheric evaporative demand around 1600 mm per year according to the FAO method (Allen et al., 1998; Jarlan et al., 2015). Soil is characterized by a fine texture with 47% of clay, 33% of loam and 18,5% of sand (Er-Raki et al., 2007). The experiment has been setting up in a rainfed wheat ("Bour") field since 2013 (Ali Eweys et al., 2017; Amazirh et al., 2018; Merlin et al., 2018). Located within a larger area occupied by rainfed wheat 'Bour'. This field was chosen to be representative at a scale of 1 km, thus enabling the comparison between 1 km resolution satellite-derived and localized in-situ measurements. The field was seeded in September 2014, September 2016 and September 2017 for the 2014-2015 (S1), 2016-2017 (S2) and 2017-2018 (S3) agricultural seasons, respectively. However, It was not ploughed(remained as bare soil) during the 2015-2016 (B1) agricultural season due to an unusual lack of precipitation in autumn-winter 2015 (Merlin et al., 2018).

The field was instrumented by an Eddy covariance (EC) system at a 2-m height. EC tower includes a CSAT3 3D sonic anemometer that measures the wind and temperature fluctuations, and a krypton hygrometer KH20 that measures the con-



centration of water vapour. The EC tower is also equipped with a CNR1 radiometer (Kipp and Zonen) to measure the four components of the net radiation (Rn) with several heat flux plates (HFT3-L, Campbell Scientific Ltd) to measure the soil heat flux (G). Energy balance closure analysis indicated that the available energy (Rn-G) was generally higher than the EC measurements. The relative closure was about 68%, 76%, 79% and 79% for S1, S2, S3 and B1, respectively. The sensible and

latent heat fluxes (H and LE) were finally corrected to force the closure of the energy balance by the Bowen's ratio method (Twine et al., 2000). LST is measured at the EC station by using two Apogee IRTS-P infrared radiometers, oriented downward and measuring the surface leaving radiance between 8 to 14 μm, set up at a 2-m height above ground. An estimate of $LST$ is obtained by averaging both measurements. The soil water content is measured at various depths (5, 10, 20, 30, 50, 70 cm) using Time Domain Reflectometry probes (model CS616) installed in a soil pit at the bottom of the EC tower. A weather station

was set up nearby the studied field to measure air temperature, solar radiation, relative humidity, wind speed and rainfall at 30-minute time step.

## 2.2 Remote sensing data

### 2.2.1 MODIS

Three products from the MODIS sensor onboard Terra and Aqua satellites are used in this study: 1) MODIS Terra/Aqua

Land Surface Temperature product ((MOD11A1/MYD11A1), 2) MODIS Terra Vegetation Indices product (MOD13A2) and 3) MODIS Albedo (combined Terra and Aqua) product (MCD43A3). All products are gridded in the Sinusoidal projection.

The MOD11A1 and MYD11A1 provide $LST$ at 1 km spatial resolution under clear-sky conditions, derived from Terra and Aqua, respectively. Brightness temperature from bands 31 and 32 are used to derive $LST$ through a generalized split-window algorithm. The MOD13A2 provides several vegetation indices at 1 km resolution. One particular vegetation index of interest

in this study is $NDVI$, available at 16-day temporal intervals. This product is derived from bands 1 and 2 of the MODIS Terra satellite.

The obtained $NDVI$ is used to derive the leaf area index (LAI) via the following formulation (Wang et al., 2013):

$$LAI = \left( NDVI \times \frac{1 + NDVI}{1 - NDVI} \right)^{1/2} \tag{1}$$

The vegetation cover fraction is expressed as (Kustas and Norman, 1997):

$$f_c = 1 - \exp(-0.5 LAI) \tag{2}$$

Finally, the MCD43A3 product provides the surface albedo ($\alpha$) at 500 m resolution every 16 days. The latter is generated from both Terra and Aqua products. In this work, the shortwave broadband $\alpha$ is used by integrating its value over the entire solar emission spectrum (0.3-5.0 $\mu$m). This value is obtained as a weighted average of the directional hemispherical reflectance (black-sky-$\alpha$) and the bi-hemispherical reflectance (white sky $\alpha$) using their two extreme values. The current $\alpha$ (called "blue

sky") is a weighted average between these two extreme cases (Lewis and Barnsley, 1994). Herein, the percentages of 85% and 15% are used for the direct and diffuse lights, respectively.



### 2.2.2 SMOS

The SMOS mission measures the natural (passive) microwave radiation around the frequency of 1.4 GHz (L-band). It aims to monitor $SM$ at a depth of about 3–5 cm with a spatial resolution of about 40 km and an accuracy better than 0.04 $m^3/m^3$ (Kerr et al., 2012). The revisiting time at the equator is 3 days for both ascending and descending passes, which are sun synchronous

at 6 am and 6 pm respectively. The SMOS level-3 1-day global $SM$ product (MIR CLF31A/D) posted on the $\sim$ 25 km Equal Area Scalable Earth (EASE) version 1.0 grid is used as input to DisPATCh algorithm.

### 2.2.3 DisPATCh

The DisPATCh remote sensing algorithm combines the coarse scale microwave-retrieved $SM$ with high-resolution optical/thermal data within a downscaling relationship to produce $SM$ at higher spatial resolution. A detailed description of the

algorithm can be found in (Merlin et al., 2012) and (Malbéteau et al., 2016). The LST is first decomposed into its soil and vegetation components, which allows the soil evaporation and plant transpiration to be estimated separately (Merlin et al., 2012). The revised soil temperature is then used to estimate the soil evaporative efficiency (SEE), which is defined as the ratio of actual to potential soil evaporation. Finally, DisPATCh converts the high-resolution optical-derived SEE fields into high- resolution $SM$ fields given a semi-empirical SEE model and a first-order Taylor series expansion around the SMOS observation. In our

application, we applied DisPATCh to 40 km resolution SMOS level-3 $SM$ and 1 km resolution MODIS optical/thermal data to produce $SM$ at a 1 km resolution (Molero et al., 2016). The input datasets are composed of: MODIS $LST$, MODIS $NDVI$ and the GTOPO Digital Elevation Model (DEM) used to correct $LST$ for topographic effects (Malbéteau et al., 2016; Merlin et al., 2013).

### 2.3 Methods

### 2.3.1 TSEB-SM

The recently developed TSEB-SM is fully described in Ait Hssaine et al. (2018b). Only the main equations are reminded below. The originality of TSEB-SM is to integrate $SM$ observations in addition to $LST$ and vegetation cover fraction data, in order to calibrate both the soil resistance to evaporation (constant parameters) and the PT coefficient on a daily basis. The model is based on the original TSEB formalism, meaning that the energy balance for vegetation is the same as in TSEB using the

PT formula, although the soil evaporation is estimated as a function of $SM$ using a soil resistance developed by Sellers et al. (1992). The use of the soil resistance formulation is justified by the fact that its main parameters ($a_{rss}$, $b_{rss}$) can be ajusted based on soil texture characteristics (Merlin et al., 2016) or by combining $SM$ and $LST$ data under bare (Merlin et al., 2018) or partially covered (Ait Hssaine et al., 2018b) soil conditions.

The vegetation latent heat flux (LEveg) is estimated via the PT formulation:

$$LE_{veg} = \alpha_{PT}.f_g.\frac{\triangle}{\triangle + \gamma}.R_{n,veg}$$  (3)





where $\alpha_{PT}$ is the PT coefficient, $f_g$ the fraction of green vegetation, $\gamma$ the psychometric constant ($\approx 67$ Pa $K^{-}1$), $\triangle$ the slope of the relationship between saturation vapor pressure and air temperature, and $R_{n,veg}$ is the vegetation net radiation. Note that, $f_g$ is set to 1 as the $\alpha_{PT}$ coefficient is varied (through the calibration procedure) to take into account the fraction of transpiring vegetation. The soil latent heat flux is estimated using the resistance formulation:

$$LE_{soil} = \frac{\rho c_p}{\gamma} \frac{e_s - e_a}{r_{ah} + r_s + r_{ss}} \tag{4}$$

where $e_s$ is the saturated vapor pressure at the soil surface, $e_a$ the actual air vapor pressure, $r_{ah}$ the aerodynamic resistance to heat transfer across the canopy–surface layer interface, $r_s$ the resistance to heat flux in the boundary layer immediately above the soil surface, and $r_{ss}$ the resistance to vapor diffusion and capillary flux in the soil. $r_{ss}$ is computed as a function of $SM$ and is expressed as (Sellers et al., 1992):

$$r_{ss} = \exp(a_{rss} - b_{rss} \times \frac{SM}{SM_{sat}}) \tag{5}$$

with $SM$ being the 0–5 cm SM, $a_{rss}$ and $b_{rss}$ are two empirical parameters (to be calibrated) and $SM_{sat}$ is the $SM$ at saturation expressed as (Cosby et al., 1984):

$$SM_{sat} = 0.1 \times (-108 \times f_{sand} + 49.305) \tag{6}$$

with $f_{sand}$ being the sand percentage of soil. In Ait Hssaine et al. (2018b), an innovative calibration approach of $\alpha_{PT}$, $a_{rss}$ and $b_{rss}$ is developed from in-situ $SM$ and $LST$ data (Ait Hssaine et al., 2018b). The $a_{rss}$ and $b_{rss}$ parameters are determined by considering that, when $f_c$ is lower than a given threshold ($f_{c,thres}$), the dynamics of total $LE$ is mainly controlled by the temporal variation of soil evaporation. Meaning that both soil parameters are estimated when the PT coefficient can be set to a constant value. Once the soil resistance has been calibrated, the PT coefficient is retrieved on a daily basis when $f_c$ is larger than $f_{c,thres}$. In fact, an iterative loop is run on soil and vegetation parameters to reach convergence of all parameters. $LST$ and $SM$ data are thus used for calibration, while the calibrated TSEB-SM is run on a daily basis using $SM$ data as forcing solely (in addition to vegetation cover fraction data). In this paper, an improvement is made on the former version of TSEB-SM to normalize the output fluxes using the LST-derived available energy. Therefore, the new version of TSEB-SM uses both $LST$ and $SM$ data (in addition to vegetation cover fraction data) as forcing on a daily basis. In practice, the latent and sensible heat fluxes derived from the TSEB-SM model are re-computed using the TSEB-SM derived evaporative fraction (EF, defined as the ratio of latent heat to available energy) and the LST-derived available energy. The rationale is that numerous modelling studies have shown the regularity and constancy of EF during daylight hours in cloud-free days (Gentine et al., 2011; Lhomme and Elguero, 1999; Shuttleworth et al., 1989) and the EF has a strong link with $SM$ availability (Bastiaanssen and Ali, 2003), which is an important factor for estimating latent heat flux. For that purpose, the $LST$ data collected at the Terra and Aqua-MODIS overpass times are used separately to estimate the instantaneous $R_n$ and G. A ratio between the daily (obtained as an average value between Aqua and Terra overpass times) latent heat flux $LE_{daily}$ and the daily available energy ($R_{n,daily}$-$G_{daily}$) is used





to calculate an average daily EF:

$$EF = \frac{LE_{daily}}{R_{n,daily} - G_{daily}} \tag{7}$$

The daily EF and the instantaneous available energy (calculated using Terra and Aqua MODIS $LST$, separately) are finally used to re-calculate the instantaneous TSEB-SM output of $LE$ and $H$ by the following formulas:

$$LE = EF \times (R_n - G) \tag{8}$$

$$H = (1 - EF) \times (R_n - G) \tag{9}$$

### 2.3.2 Uncertainty in TSEB-SM input data

The $LST$ collected by MODIS at Terra and Aqua overpass times and the $SM$ product derived at 1 km resolution from the DisPATCh algorithm applied to SMOS data, are used as input to TSEB and TSEB-SM models. Validation of TSEB and TSEB-SM input data prior to the evaluation of models output is an important issue, because of the scale discrepancy between the spatial resolution (1 km) of MODIS/DisPATCh data and the footprint of the EC flux measurements that does not exceed 100 m (Schmid, 1994).

Several studies have demonstrated the effectiveness of DisPATCh 1km resolution SM. Malbéteau et al. (2016) compared DisPATCh $SM$ data with the in-situ measurements collected in the Murrumbidgee catchment in Southeastern Australia. Their results showed that DisPATCh improved the spatial representation of $SM$ at 1 km resolution (compared to the original 40 km resolution SMOS SM), especially in semi-arid areas. Recently, Malbéteau et al. (2018) combined the DisPATCh $SM$ over the entire year 2014 (Sidi Rahal-Morocco) with the continuous predictions of a surface model in order to obtain a better estimate of daily $SM$ at 1 km resolution. They found that the assimilation of DisPATCh data improved quasi systematically the dynamics of SM.

Figure 2 shows the scatter plots of MODIS $LST$ (at Terra and Aqua overpass) versus in-situ measurements for the four agricultural seasons separately. The obtained $R^2$, $RMSE$, and $MBE$ are reported in Table 1. The statistical comparison shows strong linear correlations ($0.76 \leq R^2 \leq 0.90$) for all years. The $RMSE$ is around 4 K for S2 (2016-2017) and S3 (2017-2018) agricultural seasons while it reaches 6 K for S1 (2014-2015) and B1 (2015-2016), respectively. The observed scatter may stem from the fact that the localized (1 or 2 m wide) in-situ $LST$ is not fully representative of the 1 km resolution MODIS pixel (Ait Hssaine et al., 2018a; Yu et al., 2017). For all years (S1-3, B1), it can be seen that the $MBE$ is negative. Note that the $MBE$ is the greatest when the temperatures are largest. Such a systematic error is probably due to the non-representativeness of the in-situ $LST$ observations when compared to the corresponding scale of MODIS observations.

In order to evaluate the 1 km resolution $SM$ during the study period, Figure 3 shows a comparison of DisPATCh $SM$ with in-situ measurements for the four wheat agricultural seasons (S1, B1, S2 and S3) separately. The statistical results including the coefficient of determination ($R^2$), the root mean square error ($RMSE$), and the mean bias error ($MBE$) are reported in Table





1. The $R^2$ ranges from 0.27 to 0.55, the $RMSE$ from 0.04 to 0.09 $m^3/m^3$ and the $MBE$ from -0.05 to -0.03 $m^3/m^3$. These results are encouraging considering the heterogeneous land use composed of rainfed wheat, bare soil, fallow and farm building (see Figure 4) . In fact the localized in-situ measurements may not be perfectly representative of the 1 km resolution satellite data. Note that the efficiency of DisPATCh is supposedly higher for low $SM$ values (Malbéteau et al., 2016), which is clearly

illustrated during B1 season, while it is lower for high $SM$ values (after rain events). This can be explained by the constraints of atmospheric and vegetation conditions on disaggregation results, as well as the saturation of SEE in the higher $SMSM$ range. Another major issue that can lead to differences between DisPATCh and in-situ $SM$ is that the ground $SM$ sensors are buried at a depth of 5 cm while the penetration of the L-band wave varies between 2 and 5 cm depending on soil conditions (notably $SM$ content, texture). For S2, the $SM$ provided by DisPATCh underestimated field measurements, especially in the

higher $SM$ range. This particular behaviour could be explained by the particularly low precipitation amount during this year. Especially, it is possible that the surrounding plots were not sown by neighbour farmers, resulting in a soil that dried quickly compared to our field, which retained the $SM$ for a longer period of time.

Note that despite the relative heterogeneity within the 1 km pixel (characterized by rainfed wheat in addition to bare soil and fallow), the comparison between field measurements and 1 km resolution satellite data reflects acceptable accuracies.

## 15 3 Results and Discussion

In this section, the $a_{rss}$ and $b_{rss}$ parameters and the $\alpha_{PT}$ are firstly retrieved by following the two-step calibration based on a threshold of $f_c$ (cited in method's section). Then, the obtained calibrated values are used to estimate the surface fluxes using TSEB-SM. Finally, TSEB-SM fluxes are evaluated against the eddy covariance measurements, and results are compared with the original TSEB. To facilitate the interpretation of the simulation results using MODIS and SMOS/DisPATCh data as input,

the calibration and validation steps are previously tested using in-situ (LST and SM) data.

### 3.1 Retrieving $a_{rss}$ and $b_{rss}$ parameters

The soil resistance $r_{ss}$ is inverted for $f_c \leq f_{c,thres}$, between 11 am and 2 pm and at Terra and Aqua overpass time step for in-situ and satellite data, respectively. The result of this inversion is correlated to the actual to saturated soil moisture ratio $SM/SM_{sat}$ to determine $a_{rss}$ and $b_{rss}$ parameters. The calibration process is applied for each season independently. Then a

pair ($a_{rss}$, $b_{rss}$) is calculated for the entire study period for in-situ and satellite data, respectively.

Figure 5a and 5b plot the $\log(r_{ss})$ versus in-situ $SM/SM_{sat}$ using in-situ and satellite data, respectively. The mean retrieved values (7.62, 2.43) and (7.32, 4.58) for in-situ and satellite data, respectively, are close to the values found in Sellers et al. (1992) (8.2, 4.3) and in Ait Hssaine et al. (2018b) (7.2, 4). However, by comparing both figures (5a and 5b), one notes that the use of in-situ data generates more scatter than with satellite data. The apparent scatter in retrieved $r_{ss}$ could be interpreted by

the impact of the daily cycle of meteorological (evaporative demand) conditions or soil properties differences (Merlin et al., 2011, 2016, 2018). The retrieved soil parameters also vary from year to year: the standard deviation is 0.39 and 1.69 for $a_{rss}$ and $b_{rss}$, respectively. This can be explained by the compensation effects linking $a_{rss}$ and $b_{rss}$ parameters which prove the



empirical nature of the $r_{ss}$. Another major issue that can lead to these differences is the depth of $SM$ measurements (Merlin et al., 2011). In Sellers et al. (1992), the near-surface soil moisture is defined in the 0-5-cm soil layer, whereas in our field, $SM$ measurements are made at 5-cm depth. Also, the sensing depth of SMOS observations is generally shallower than the in-situ surface measurements (Escorihuela et al., 2010). Moreover, the variability of $a_{rss}$ and $b_{rss}$ in Figure 5b using remote sensing

data can be linked to the scale difference between DisPATCh SM/MODIS products (1 km) and the field measurements. As shown in Figure 3, the field is surrounded by trees, buildings and fallows, which causes the spatial heterogeneity within the pixel of 1 km. This heterogeneity can introduce errors on the model inversion. Nevertheless, soil parameters are quite similar for in-situ and satellite data sets. Therefore, the heterogeneity issues within the 1 km pixel scale are minor in this study.

## 3.2 Time series of daily retrieved $\alpha_{PT}$

The second calibration step consists in inverting the daily $\alpha_{PT}$ when vegetation is covering a significant part of soil ($f_c > f_{c,thres}$), for the three seasons of rainfed wheat (S1, S2 and S3), by using in-situ data and satellite data, separately. Herein, the calibration of $\alpha_{PT}$ is bounded by a minimum (0) and maximum (2) acceptable physical value, in order to avoid unacceptable values of $\alpha_{PT}$ that can be produced because of the uncertainties in daily LST estimates. Such an upper bounding is especially needed when vegetation partially covers the soil.

### 3.2.1 Using in-situ data

Figure 6 plots the daily variation of $\alpha_{PT}$ for each season (S1, S2 and S3) separately, using in-situ data. The mean retrieved values of $\alpha_{PT}$ are 1.26, 1.12 and 1.09 for S1, S2 and S3 respectively. In all cases, the mean $\alpha_{PT}$ is close to the theoretical $\alpha_{PT}$ value (1.26). It is well observed that the retrieved $\alpha_{PT}$ for S1 is slightly larger compared to those obtained for both S2 and S3. This can be explained by the timing and amount of rainfall during each season. Note that unexpected low values of $\alpha_{PT}$

are recorded for S3 during the first few days (25 January-4 March) of the development stage. They may be associated with uncertainties in retrieved $\alpha_{PT}$ as the impact of soil surface is still significant, as well as to a relatively low evaporative demand especially since this period coincides with cloudy days and abundant precipitations. Indeed, the coupling between transpiration (and hence retrieved $\alpha_{PT}$) and $LST$ is expected to be lower under lower atmospheric demand.

The retrieved $\alpha_{PT}$ is then smoothed as in Ait Hssaine et al. (2018b) to remove outliers and to reduce uncertainties at the

daily time scale. The smoothed values of $\alpha_{PT}$ range from 0 to 1.54, 0 to 1.38 and 0.45 to 1.43 for S1, S2 and S3 respectively. The maximum of $\alpha_{PT}$ is close to 1.26 for S2, while it is higher for S1 and S3. This result is in accordance with the total rainfall amounts which were about 608, 214 and 421 mm for S1, S2 and S3 respectively. Additionally, one can state that the stability of $\alpha_{PT}$ strongly depends on the rainfall distribution along the agricultural season. The daily $\alpha_{PT}$ is more stable for S1 than for S2 and S3. Indeed, the amount of rain during S1 is very important with two peaks of about 83 mm that occurred at the beginning of the season and during the growing stage. The second one coincides exactly with the maximum value of the retrieved $\alpha_{PT}$.

However, different results are obtained for S2 compared to S1 due to the lowest precipitation amount recorded over that season. As shown in Figure 6 the amount of rain is concentrated at the beginning of the growing stage (mid December), when the $\alpha_{PT}$ peaks. Afterward, the smoothed $\alpha_{PT}$ tends to decrease because of insufficient soil water reserve in the root zone to enable





wheat to continue growing. Rainfall is also significant for S3 and every rainfall event causes an immediate (daily) response of $\alpha_{PT}$ (after 4th March). As mentioned before, the significant error in $\alpha_{PT}$ retrievals for S3 between 25 January and 4 March induces strong uncertainties in the smoothing function estimates.

### 3.2.2 Using satellite data

Figure 6 illustrates also the daily variation of $\alpha_{PT}$ retrieved from satellite data for each season separately. S1 and S2 have a very similar distribution of the retrieved $\alpha_{PT}$ as compared to the retrieved $\alpha_{PT}$ using in-situ data, respectively. For S3, only six retrieved $\alpha_{PT}$ values are available because of the non-availability of MODIS products during cloudy days. For this reason, no information linked to the variability of $\alpha_{PT}$ can be derived during this season. The retrieved values are smoothed and superimposed with the rainfall events. It is clearly shown that the smoothed $\alpha_{PT}$ for S1 and S2 have the same shape with a

small variability, when comparing with the smoothed $\alpha_{PT}$ using in-situ data, resulting in an error estimated as the $RMSE$ to the mean $\alpha_{PT}$ ratio, of about 11 and 19 %, for S1 and S2 respectively. For S1 the maximum of smoothed $\alpha_{PT}$ is reached at the same time as when using the in-situ data, with a value of about 1.38, while the maximum for S2 is reached 10 days before the maximum of the $\alpha_{PT}$ derived from in-situ data with little response of $\alpha_{PT}$ to rainfall events. These differences may be linked to uncertainties in disaggregated SMOS SM, as well as to the weaker availability of satellite data. Because of the small number

of data points (retrieved $\alpha_{PT}$) during S3, the smoothed $\alpha_{PT}$ remains at a mostly constant value ($\sim 0.7$) throughout the study period, with a significant relative difference of about 34 % when comparing with the $\alpha_{PT}$ retrieved using in-situ data.

### 3.2.3 Interpretation of $\alpha_{PT}$ variabilities

Figure 7 plots variation of calibrated daily $\alpha_{PT}$, superimposed with $NDVI$ and rainfall events. It is visible that the maximum value of $NDVI$ appears sooner than the maximum value of $\alpha_{PT}$ for both S1 and S3. Such a delay is attributed to the high

soil moisture level in the root-zone during the maturity stage. Later in the season, $\alpha_{PT}$ decreases as $NDVI$ starts to decline at the onset of senescence. In contrast, the maximum value of $NDVI$ appears later than the maximum value of $\alpha_{PT}$ for S2. This can be explained by the fact that rainfall at the beginning of the development phase satisfies the plant requirements, while the rainfall amount during the development stage is relatively low compared to the crop water needs (Kharrou et al., 2011). Large variations in $\alpha_{PT}$ occur during the agricultural season, as a result of the amount, frequency, and distribution of rainfall

along the season. In general, the analysis of the $\alpha_{PT}$ variability using satellite data illustrates the robustness of the proposed approach, which combines microwave and optical/thermal data to retrieve a water stress indicator at the daily time scale.

### 3.3 Surface fluxes

The robustness of TSEB and TSEB-SM for partitioning $(R_n - G)$ into $H$ and $LE$ is evaluated using in-situ and remotely sensed LST, SM, NDVI, separately, at Terra and Aqua MODIS overpasses.





### 3.3.1 Using in-situ data

Figure 8 shows an intercomparison of simulated and observed $LE$ for the 4 seasons separately. TSEB-SM clearly provides improved results compared to the original TSEB. The obtained values of $RMSE$ by TSEB-SM are about 68 and 72 W/$m^2$ for S1 and S2 respectively, which is significantly lower than those revealed by TSEB (109 and 86 W/$m^2$, respectively) (see Table

2). For B1 (season of bare soil), TSEB largely overestimates $LE$ with a $MBE$ of about 165 W/$m^2$ compared to TSEB-SM, which yields a $MBE$ of 59 W/$m^2$. This overestimation of TSEB is most probably related to an inadequate value of $\alpha_{PT}$ (=1.26) for bare soil surfaces. In fact, 1.26 is an optimum value for the potential transpiration rate (Agam et al., 2010; Chirouze et al., 2014). In the case of TSEB-SM, biases are reduced thanks to the calibration of the $r_{ss}$ resistance. Additionally, according to TSEB-SM assumptions, $\alpha_{PT}$ for $f_c \leq 0.5$ is set to the average value of the $\alpha_{PT}$ retrieved for $f_c > 0.5$. During B1 season

(bare soil conditions), $\alpha_{PT}$ was hence obtained as an average value of the mean $\alpha_{PT}$ retrieved for all seasons S1, S2 and S3 when $f_c > 0.5$ ( $\alpha_{PT} \sim 1$). However, this value remains relatively high for a bare soil, which yields a slight overestimate of $LE$ measurements (see B1 case in Figure 8).

For S3 season, the error on daily retrieved $\alpha_{PT}$ at the beginning of the development stage has a strong impact on $LE$ predictions and thus yields to greater discrepancies illustrated in Figure 8. To overcome this error, the threshold on $f_c$ to

separate calibration steps 1 and 2 was increased to 0.63 (arbitrary value). The TSEB-SM model is then run using the new threshold. The $LE$ simulations are improved, with a $RMSE$ of 73 W/$m^2$ instead of 98 W/$m^2$ and a relative error (estimated as the $RMSE$ divided by the mean observed LE) of about 42 % instead of 58 %. The increase in the threshold is intended to decrease the uncertainties in $\alpha_{PT}$ retrievals when vegetation is not fully covering the soil. It can be concluded that the errors in $\alpha_{PT}$ retrievals have a strong impact on $LE$ estimates.

The ability of TSEB-SM to estimate the sensible heat fluxes is also investigated. Figure 9 displays the comparison between TSEB and TSEB-SM for each season and Table 2 2 summarized the different statistical parameters. One can notice that TSEB shows greater discrepancies in $H$ estimation, with a $RMSE$ of about 127, 112 and 103 W/$m^2$ and $MBE$ of about -41, 1, and -71 W/$m^2$ for S1, S2 and S3 respectively. Both $RMSE$ and $MBE$ values are generally much reduced when using TSEB-SM with $RMSE$ values of about 68, 72, and 98 W/$m^2$ and MBE$MBE$ values of about -10, 24, and 7 W/$m^2$, respectively. During

B1, TSEB model underestimates H. This can be explained by the low-sensitivity of simulated sensible heat flux to changes in surface and atmospheric conditions, consistent with former results obtained on a different sites of irrigated wheat (Ait Hssaine et al., 2018b). The discrepancies between TSEB-SM and in-situ $H$ during S3 are mostly rectified by using the new threshold on $f_c$: the statistical results are improved, the $RMSE$ is about 73 W/$m^2$ and the relative error is 39 % (instead of 52 %). It can be concluded that the uncertainty observed over the $\alpha_{PT}$ during the first few days of development stage (25 January-4 March)

is mainly related to the impact of the soil, which is not negligible during the first weeks of the growing stage. Nevertheless, by considering the overall results obtained for the 3 seasons, the threshold of $f_{c,thres} = 0.5$ can be considered as an acceptable value to calibrate the soil resistance parameters and the Priestly Taylor coefficient.

As a further step, the intercomparison between TSEB and TSEB-SM is evaluated by predicting $R_n$ and $G$ fluxes instead of forcing them to their measured values. The statistical results of the comparison between simulated and observed $R_n$, G,$H$ and





$LE$ are listed in Table 3. The scattering obtained when comparing turbulent flux estimations to measurements is mainly related to the uncertainty in available energy estimates, mainly related to the uncertainty in soil heat flux estimates. Indeed, as reported in Table 3, $R_n$ is very well simulated for both TSEB and TSEB-SM. The $R^2$ between simulated and observed $R_n$ is about 0.99 during all seasons. Meanwhile $G$ shows a poor correlation, with an $R^2$ varying from 0.05 to 0.45. This is mainly linked to
the approach used to estimate G, which requires local calibration. Kustas et al. (1998) hence indicated that the ratio G/$R_{n,soil}$ cannot be considered as a constant, because it is affected by different factors such as time of day, moisture conditions and soil texture and structure.

### 3.3.2   Using satellite data

In order to gain greater insight into how TSEB and TSEB-SM models respond to different surface conditions across a landscape,
an analysis of the spatial distributions and the magnitude of the turbulent fluxes using remotely sensing data produced from the two models is conducted. The comparisons between TSEB/TSEB-SM versus observed $LE$ over the four seasons are illustrated in Figure 10.Figure 10 indicates that TSEB overestimates latent heat flux. The overall $MBE$ are about 119, 181, 94 and 128 W/$m^2$ for S1, B1, S2 and S3 respectively. The overestimation of $LE$ fluxes can be explained by the fact that $\alpha_{PT}$ is set to 1.26 during the entire agricultural season including stress conditions. This probably causes larger errors on the $LE$ estimation
especially during the growing stage. Indeed, the saturation of TSEB during the senescence period is precisely caused by the PT coefficient fixed to 1.26. The errors are reduced when using TSEB-SM. In fact, the constraint on plant transpiration, while retrieving daily $\alpha_{PT}$ values improves ET estimates especially for the growing stage. Moreover, during the senescence stage the large positive bias of $LE$ is considerably reduced. In fact, the decrease in calibrated daily $\alpha_{PT}$ is associated with the drop in NDVI during senescence (Ait Hssaine et al., 2018b). Additionally, the constraint on the soil evaporation via the DisPATCh
SM, clearly reduces the $MBE$ values during the emergence period ($f_c \leq f_{c,thres}$). Finally, the constraint applied on TSEB-SM output fluxes using LST-derived available energy and TSEB-SM-derived evaporative fraction (Equation 8) improves the $LE$ estimates for the whole study period. The $MBE$ are about 39, 4, 7 and 62 W/$m^2$ for S1, S2, S3 and B1 respectively.

TSEB consistently exhibits larger errors on $H$ estimation (see Figure 11), with $RMSE$ values up to 98, 73, 56 and 66 W/$m^2$ during S1, S2, S3 and B1 respectively. The $RMSE$ is improved while using TSEB-SM, with values of about 55, 41, 24 and
27 W/$m^2$ during S1, S2, S3 and B1 respectively.

The intercomparison between TSEB and TSEB-SM is made by forcing the available energy to its measured value. The statistics listed in Table 3 indicate that there are similar differences between modeled versus measured $R_n$ using either TSEB or TSEB-SM. Overall, the discrepancies between estimated and measured $R_n$ are likely due to a greater scatter between MODIS and in-situ measured LST. Note that $RMSE$ values up to 6 K have been noted when comparing $LST$ MODIS with ground-
based measurements. These uncertainties are likely to be explained by the huge scale mismatch between the 1 km resolution of MODIS $LST$ and the footprint size (approximately 1 m) of ground-based radiometers. The uncertainties in key input data generate large differences in simulated $R_n$ compared to the tower measurements. The greater scatter between modeled and measured $G$ from the two models reflect the fact that there is a major mismatch in scale between the area sampled by the soil heat flux sensors and the 1 km resolution of model inputs. It appears that the $LE$ estimates from TSEB-SM are generally in





closer agreement with the measurements than the TSEB model outputs. The $RMSE$ is significantly improved from 103 to 52 W/$m^2$, from 151 to 30 W/$m^2$, from 101 to 35 W/$m^2$ and from 83 to 24 W/$m^2$, during S1, B1, S2 and S3, respectively. For the sensible heat flux H, the difference between TSEB estimates and EC measurements listed in Table 3 indicates a fairly large underestimation, the $MBE$ values varying between -56 W/$m^2$ and -240 W/$m^2$. However, the TSEB-SM output provides a quite significant improvement, with an absolute $MBE$ lower than -61 W/$m^2$ during all seasons.

## 4 Conclusions

The microwave-derived near-surface soil moisture (SM) from SMOS and thermal-derived land surface temperature (LST) from MODIS are integrated simultaneously within a calibration procedure to invert both the soil resistance to evaporation (constant parameters) and the PT coefficient based on a threshold on $f_c$. The TSEB-SM model is applied during a four-year period (2014-2018) over a rainfed wheat field in the Tensift basin, central Morocco. The first calibration step with $f_c \leq f_{c,thres}$ consists in inverting $r_{ss}$ at Terra and Aqua overpass times. Despite the scale difference between the MODIS/DisPATCh resolution data and the footprint size of in-situ measurements, the pair parameters ($a_{rss}$,$b_{rss}$) calculated for the entire study period using satellite data are relatively close to those derived from in-situ measurements. The second calibration step consists in estimating $\alpha_{PT}$ on a daily basis for $f_c > f_{c,thres}$ by using $LST$ and $SM$ data. The maximum of daily calibrated $\alpha_{PT}$ are 1.38, 1.25 and 0.87, when using satellite data, for S1, S2 and S3, respectively. Those values are in accordance with the total rainfall amounts, which were about 608, 214 and 421mm/wheat season for S1, S2 and S3 respectively. S1 and S2 have the same distribution of daily calibrated $\alpha_{PT}$ when comparing with the $\alpha_{PT}$ retrieved using in-situ data, while the retrieved $\alpha_{PT}$ remains at a mostly constant value ($\sim 0.7$) throughout the study period S3 because of the non-availability of MODIS products during cloudy days.

An analysis of the spatial distributions and the magnitude of the turbulent fluxes using remotely sensing data produced from the two models were conducted. TSEB exhibits larger errors on $H$ and $LE$ estimates. These uncertainties can be linked to the theoretical value of $\alpha_{PT}$, which is fixed to 1.26 for the whole study period, as well as to the scale mismatch between the 1 km resolution of MODIS $LST$ and the footprint size (approximately 1 m) of the ground-based radiometer. The constraint applied on the soil evaporation represented explicitly as a function of $SM$ via a soil resistance term reduces the errors when using TSEB-SM. In fact, the use of the $SM$ derived from microwave data is one of the main controlling factors of the evaporative fraction, which helps to determine with more accuracy the LE/H partitioning.

Last but not least, the coupling of the soil resistance formulation with the TSEB formalism improves the estimation of soil evaporation, and should, as a consequence, improve the partitioning of evapotranspiration. As a short term prospect, the robustness of TSEB-SM in terms of evaporation/transpiration partitioning will be tested by using independent flux measurements derived from lysimeters, and sap flow sensors, and vapour chambers (Rafi et al., 2019).

*Acknowledgements.* The study was carried out in the framework of the Joint International Laboratory TREMA https://www.lmi-trema.ma, and funded by the European Commission Horizon 2020 Programme for Research and Innovation (H2020) in the context of the Marie





Sklodowska-Curie Research and Innovation Staff Exchange (RISE) action (REC project, grant agreement no: 645642). Additional funding was provided by the ERANETMED03-62 CHAAMS, the French Agence Nationale de la Recherche (MIXMOD-E project. ANR-13-JS06-003-01) and PHC TBK/18/61 projects. We deeply grateful to the ARTS fellowship program from Institut de Recherche pour le Développement (IRD) for awarding a PhD scholarship to Bouchra AIT HSSAINE.



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





**Figure 1.** Location of Sidi Rahal site (east of Marrakech) in the Tensift basin, central of Morocco.



**Figure 2.** Scatter plots of MODIS versus in-situ $LST$ at Sidi Rahal site for S1 (2014-2015), B1 (2015-2016), S2 (2016-2017) and S3 (2017-2018) agricultural seasons, separately, (red dashed line is the line(1:1)-black line is the regression line).





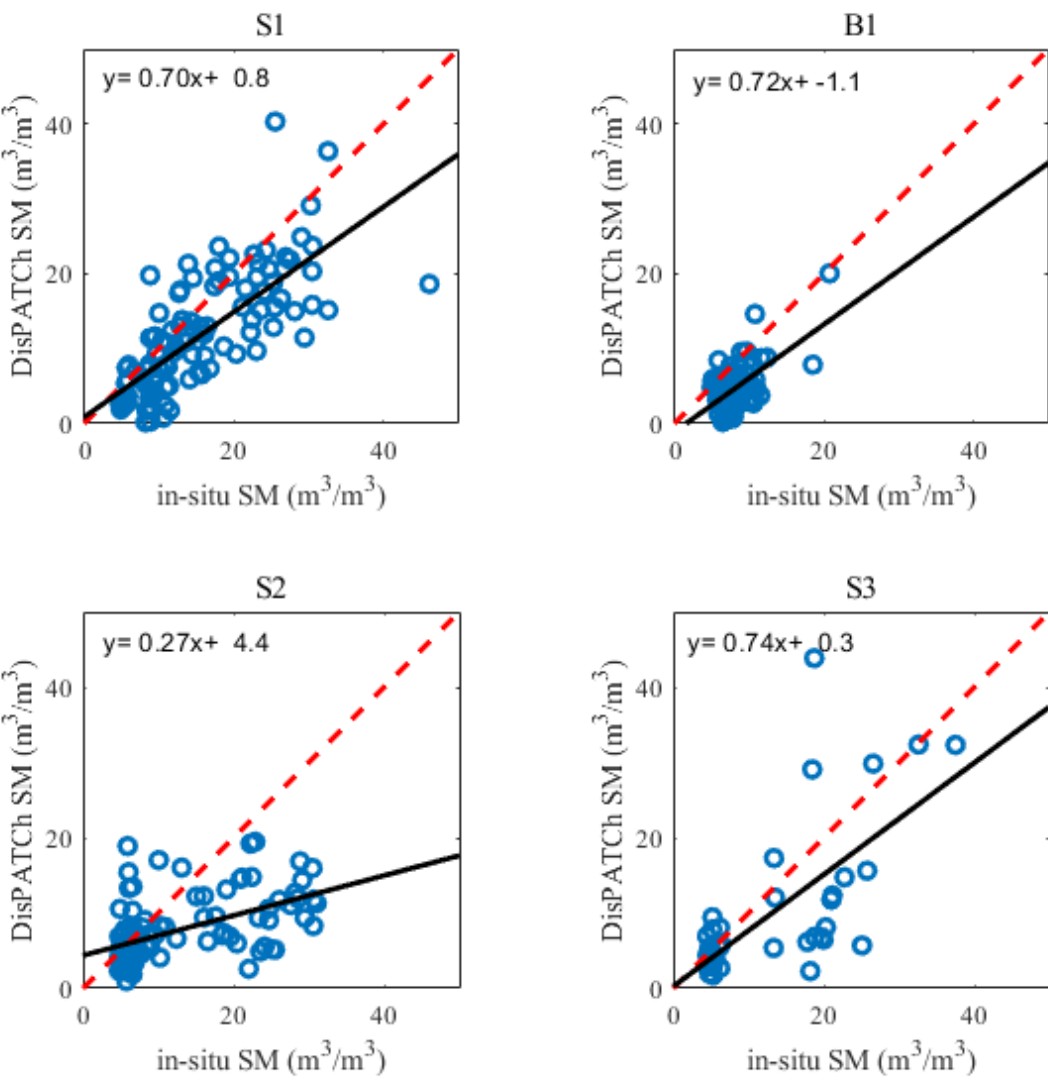

**Figure 3.** Scatter plots of the 1 km resolution DisPATCh versus in-situ $SM$ at Sidi Rahal site for S1 (2014-2015), B1 (2015-2016), S2 (2016-2017) and S3 (2017-2018) agricultural seasons, separately.





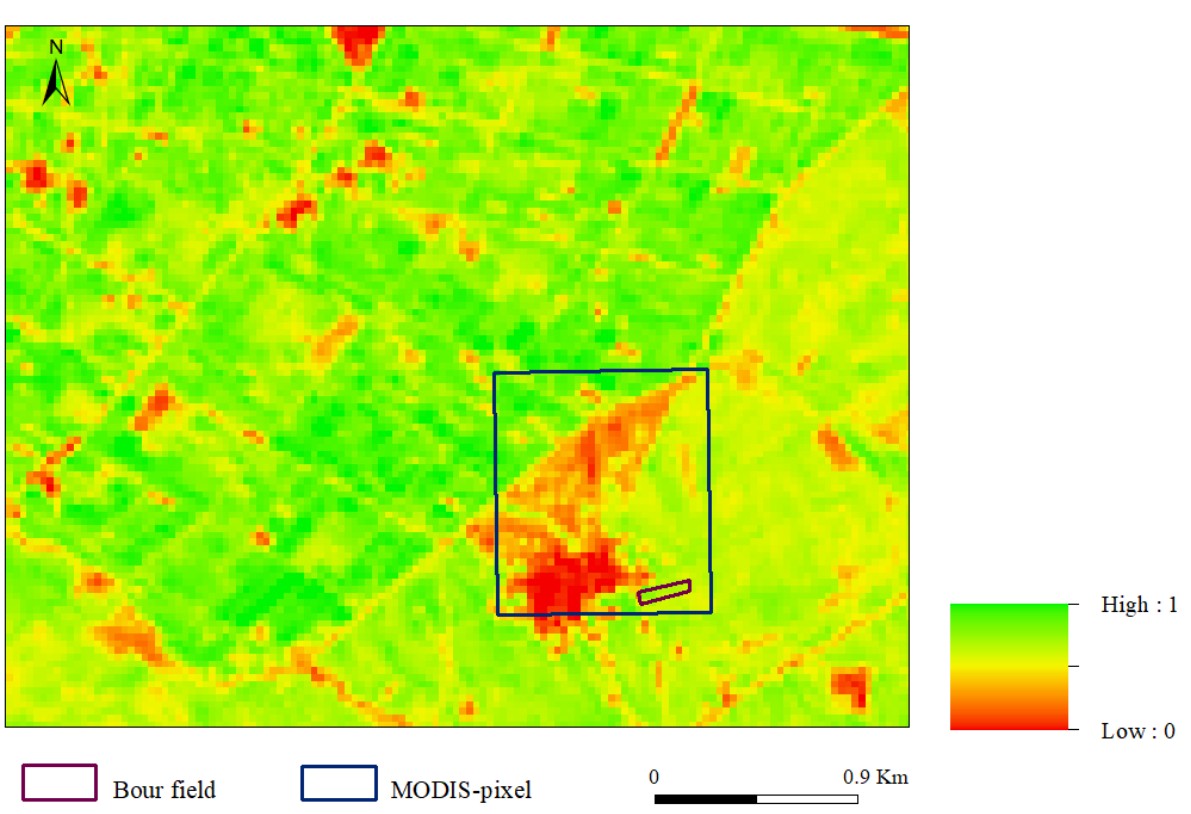

**Figure 4.** $NDVI$ image derived from Landsat data acquired on 17/04/2018. The experimental field and the overlaying 1 km resolution
MODIS pixel are superimposed.





**Figure 5.** $\log(r_{ss})$ versus $SM/SM_{sat}$ (calibration step 1) using in-situ (a) and satellite (b) data.



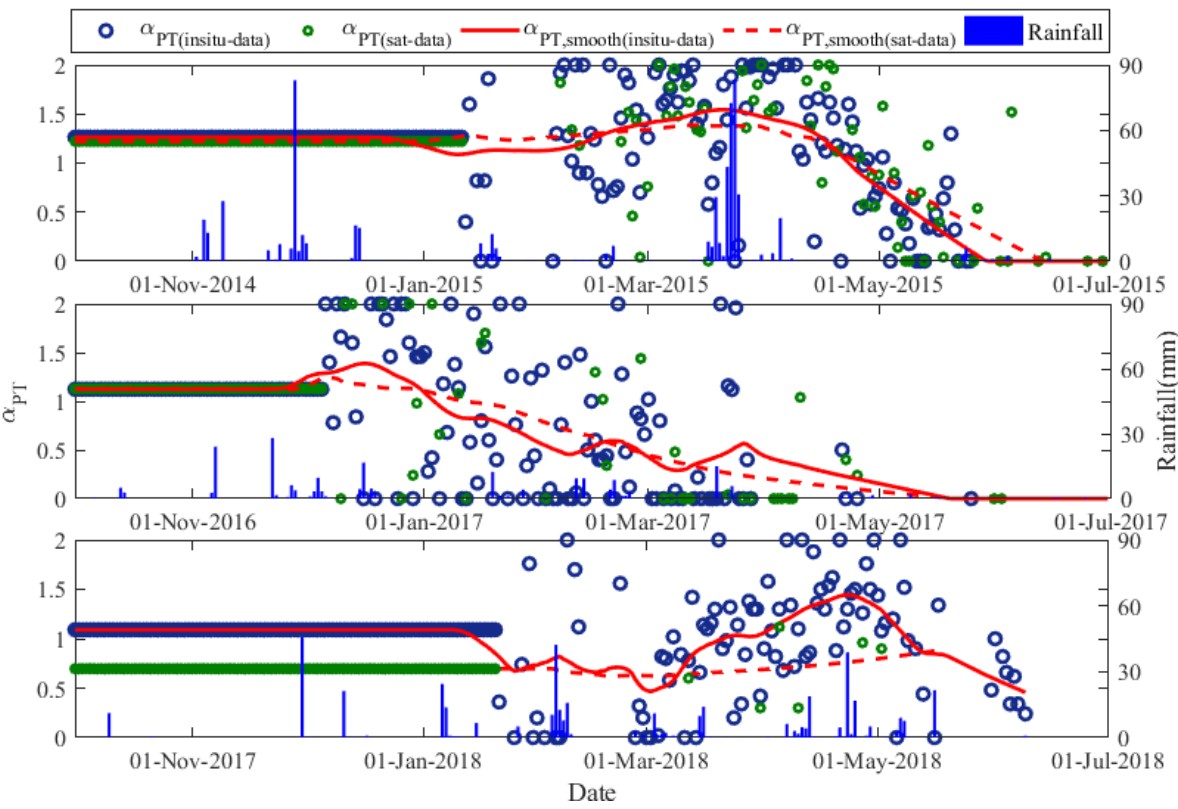

**Figure 6.** Time series of daily retrieved and smoothed $\alpha_{PT}$ (calibration step 2-using in-situ data, and satellite data) collected during S1, S2 and S3.





**Figure 7.** Time series of calibrated daily $\alpha_{PT}$ (red-using in-situ data, green- using satellite data) superimposed with $NDVI$ and the rainfall events during S1, S2 and S3, separately.





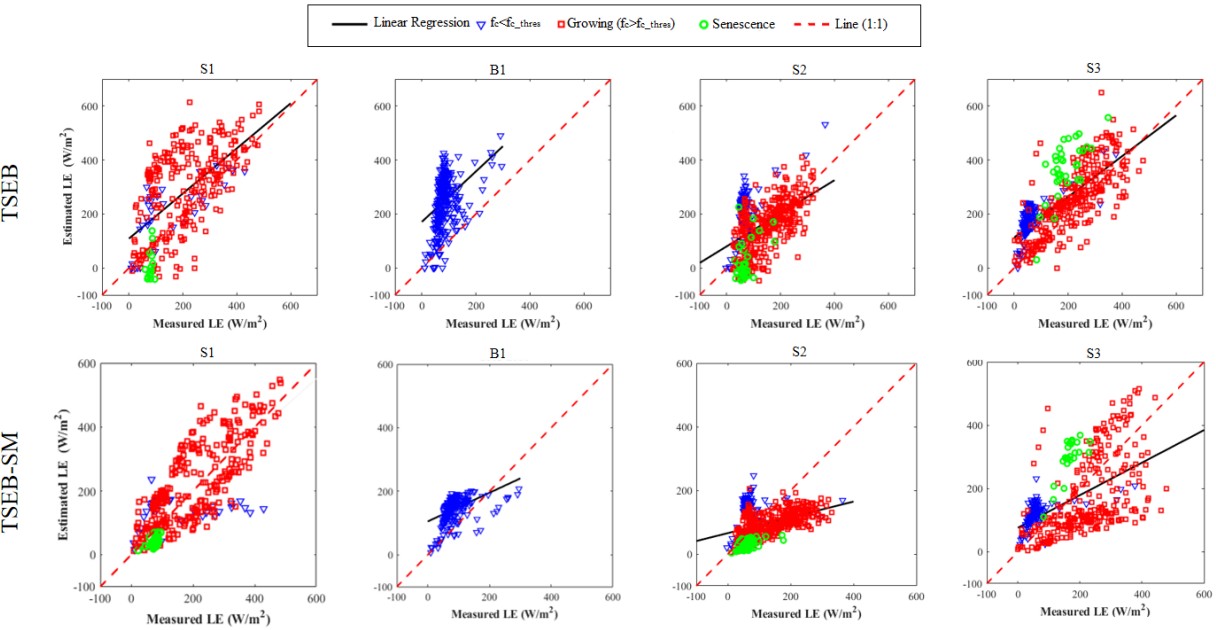

**Figure 8.** Scatterplot of simulated versus observed $LE$ for (top) TSEB and (bottom) TSEB-SM models using in-situ data collected during S1, B1, S2 and S3, respectively.



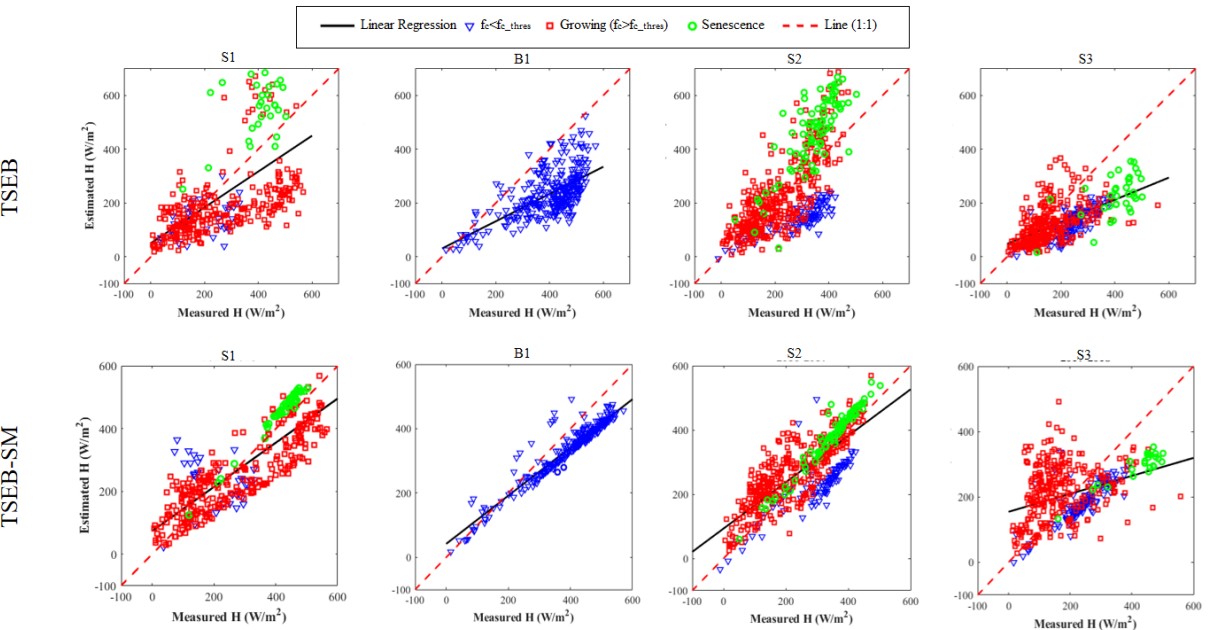

**Figure 9.** Same as Fig. 8 but for $H$ fluxes.





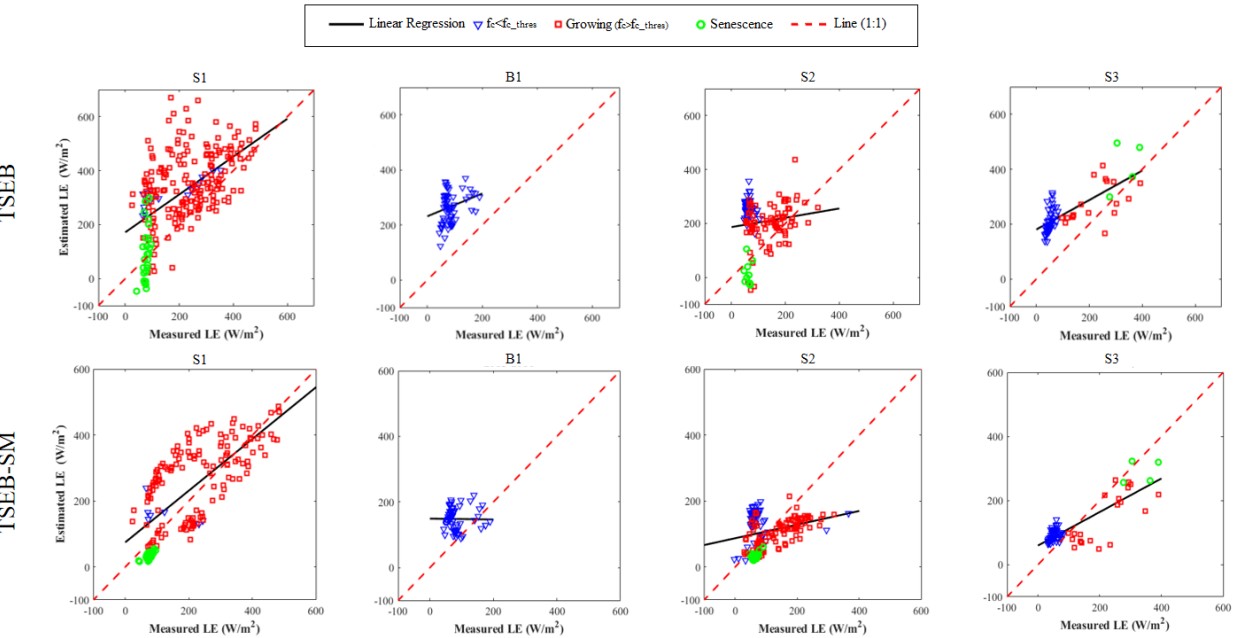

**Figure 10.** Same as Fig. 8 but for satellite data.





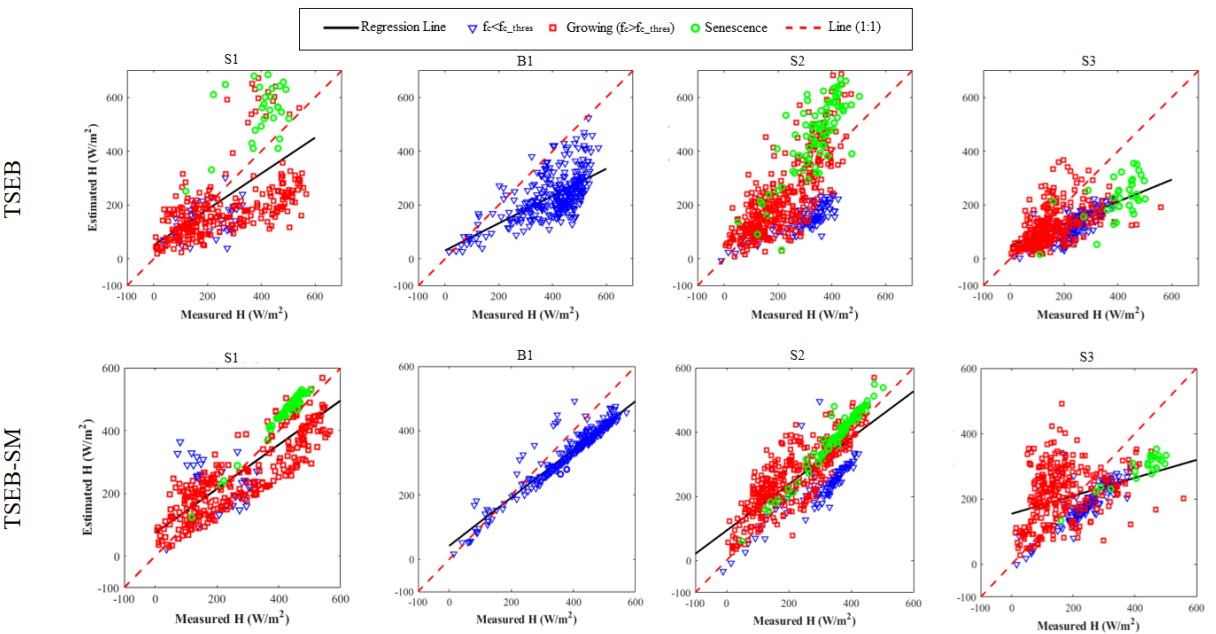

**Figure 11.** Same as Fig. 8 but for *H* fluxes and satellite data.





**Table 1.** Validation results of DisPATCh $SM$ and MODIS $LST$ at Sidi Rahal site.

| | Period | $R^2$ | $RMSE$ | $MBE$ |
|---|---|---|---|---|
| LST | S1 | 0.8 | 6.4 (K) | -3.7 (K) |
| | B1 | 0.76 | 5.6 (K) | -4.6(K) |
| | S2 | 0.91 | 4.3 (K) | -2.9 (K) |
| | S3 | 0.89 | 4 (K) | -2 (K) |
| SM | S1 | 0.55 | 0.07 $m^3/m^3$ | -0.04 $m^3/m^3$ |
| | B1 | 0.36 | 0.04 $m^3/m^3$ | -0.03 $m^3/m^3$ |
| | S2 | 0.27 | 0.09 $m^3/m^3$ | -0.05 $m^3/m^3$ |
| | S3 | 0.47 | 0.08 $m^3/m^3$ | -0.03 $m^3/m^3$ |





**Table 2.** Statistical results ($RMSE$, $R^2$ and $MBE$) between modeled and measured sensible and latent heat fluxes for S1, S2, B1 and S3, and for TSEB and TSEB-SM model, separately ($R_n$ and $G$ are forced to their measured value).

| | | | TSEB | | | TSEB-SM | | |
|---|---|---|---|---|---|---|---|---|
| | | | $RMSE$ | $R^2$ | $MBE$ | $RMSE$ | $R^2$ | $MBE$ |
| Using in-situ data | $LE$ (W/$m^2$) | S1 | 109 | 0.39 | 76 | **68** | **0.59** | **10** |
| | | B1 | 136 | 0.15 | 165 | **52** | **0.22** | **59** |
| | | S2 | 86 | **0.22** | 30 | 72 | 0.16 | -24 |
| | | S3 | 103 | **0.53** | 71 | 98 | 0.29 | **-7** |
| | $H$ (W/$m^2$) | S1 | 127 | 0.33 | -41 | **68** | **0.7** | **-10** |
| | | B1 | 136 | 0.44 | -165 | **52** | **0.91** | **-59** |
| | | S2 | 112 | 0.47 | 1 | **72** | **0.63** | **24** |
| | | S3 | 103 | **0.38** | -71 | 98 | 0.14 | **7** |
| Using satellite data | $LE$ (W/$m^2$) | S1 | 95 | 0.34 | 119 | **55** | **0.51** | **39** |
| | | B1 | 66 | **0.07** | 181 | **27** | 0.01 | **62** |
| | | S2 | 67 | 0.02 | 94 | **41** | **0.08** | **4** |
| | | S3 | 56 | 0.55 | 128 | **24** | **0.68** | **7** |
| | $H$ (W/$m^2$) | S1 | 98 | 0.3 | -104 | **55** | **0.54** | **-39** |
| | | B1 | 66 | 0.37 | -181 | **27** | **0.52** | **-62** |
| | | S2 | 73 | 0.33 | -71 | **41** | **0.6** | **-4** |
| | | S3 | 56 | 0.28 | -128 | **24** | **0.36** | **-7** |





**Table 3.** Same as Table 2 but for simulated $R_n$ and $G$.

| | | | TSEB | | | TSEB-SM | |
|---|---|---|---|---|---|---|---|
| | | $RMSE$ | $R^2$ | $MBE$ | $RMSE$ | $R^2$ | $MBE$ |
| | S1 | 35 | 0.99 | -38 | 35 | 0.99 | -38 |
| $Rn$ (W/$m^2$) | B1 | 14 | 0.99 | 12 | 14 | 0.99 | 12 |
| | S2 | 20 | 0.99 | 9 | 20 | 0.99 | 9 |
| | S3 | **7** | 0.99 | -0.46 | 7 | 0.99 | -0.46 |
| | S1 | 19 | 0.32 | 17 | 19 | 0.32 | 17 |
| $G$ (W/$m^2$) | B1 | 19 | 0.05 | 12 | 19 | 0.05 | 12 |
| | S2 | 30 | 0.28 | -13 | 30 | 0.28 | -13 |
| | S3 | 26 | 0.44 | 9 | 26 | 0.44 | 8 |
| Using in-situ data | S1 | 87 | 0.35 | 27 | **65** | **0.58** | **-21** |
| $LE$ (W/$m^2$) | B1 | 141 | 0.12 | 174 | **52** | **0.16** | 60 |
| | S2 | 91 | **0.23** | 35 | 68 | 0.22 | **-15** |
| | S3 | 91 | **0.62** | 54 | **84** | 0.47 | **22** |
| | S1 | 127 | 0.33 | -44 | **70** | **0.73** | **34** |
| $H$ (W/$m^2$) | B1 | 145 | 0.43 | -177 | **52** | **0.9** | **-60** |
| | S2 | 112 | 0.48 | **2** | **78** | **0.68** | 36 |
| | S3 | 99 | **0.3** | -64 | **87** | 0.3 | **-32** |
| | S1 | 23 | 0.94 | 8 | 22 | 0.93 | 8 |
| $Rn$ (W/$m^2$) | B1 | 85 | 0.47 | 32 | 85 | 0.47 | 32 |
| | S2 | 22 | 0.94 | 12 | 22 | 0.94 | 12 |
| | S3 | 17 | 0.97 | 2 | 17 | 0.97 | 2 |
| | S1 | 20 | 0.41 | 24 | 19 | 0.4 | 24 |
| $G$ (W/$m^2$) | B1 | 20 | 0 | 15 | 20 | 0 | 15 |
| | S2 | 25 | 0.12 | -15 | 25 | 0.12 | -15 |
| | S3 | 22 | 0.08 | 10 | 22 | 0.08 | 10 |
| Using satellite data | S1 | 103 | 0.24 | 86 | **52** | **0.49** | **28** |
| $LE$ (W/$m^2$) | B1 | 151 | **0.02** | 240 | **30** | 0.01 | **65** |
| | S2 | 101 | **0.07** | 96 | **37** | 0.06 | **28** |
| | S3 | 83 | 0.47 | 74 | **24** | **0.69** | **14** |
| | S1 | 112 | 0.34 | -91 | **63** | **0.44** | **-45** |
| $H$ (W/$m^2$) | B1 | 150 | 0.16 | -240 | **28** | **0.49** | **-61** |
| | S2 | 97 | 0.4 | -56 | **38** | **0.52** | **-4** |
| | S3 | 85 | 0.12 | -83 | **27** | **0.28** | **-29** |