# Peer review of "An evapotranspiration model self-calibrated from remotely sensed surface soil moisture, land surface temperature and vegetation cover fraction: application to disaggregated SMOS and MODIS data"

_Hydrology and Earth System Sciences, 2019_

## Referee Comment (RC1) · Anonymous Referee #1 · 3 Jun 2019

The paper titled 'An evapotranspiration model self-calibrated from remotely sensed surface soil moisture, land surface temperature and vegetation cover fraction: application to disaggregated SMOS and MODIS data' by Ait Hssaine aimed to used LST and disaggregated soil moisture to better constrain the soil evaporation of TSEB model. This is a good idea; however, the presentation of the manuscript needs substantial improvement before being published in HESS. Here are my suggestions and comments, which needs to be considered before being approved for publications. (1) The abstract is

poorly written and does not give a clear message about the novelty of the work. Re-work is necessary. (2) Introduction: The flow should be logical. Since the objective of the manuscript is to improve the soil evaporation in TSEB to meet up field-scale ET mapping challenges, I do not see any need of line 5 – 10 in page 2. (3) Introduction: 'Evapotranspiration (ET) is a crucial water flux in semi-arid areas'; should be supported by recent literature. The authors should be aware about some recently published ET modeling and mapping studies that particularly addressed the challenges semi-arid and arid ecosystems (that deserves to be cited here); for example,

Mallick et al. (2015). Reintroducing radiometric surface temperature into the Penman-Monteith formulation, Water Resources Research, 51, 6214–6243, http://doi.org/10.1002/2014WR016106. Mallick et al. (2014). A surface temperature initiated closure (STIC) for surface energy balance fluxes, Remote Sensing of Environment, 141, 243 - 261. Bhattarai et al. (2019). An automated multi-model evapotranspiration mapping framework using remote sensing and reanalysis data. Remote Sensing of Environment, 229, 69 - 92. Gerhards et al. (2019). Challenges and Future Perspectives of Multi-/Hyperspectral Thermal Remote Sensing for Crop Water Stress Detection: A Review, Remote Sensing, 11(10), 1240; https://doi.org/10.3390/rs11101240. Bhattarai et al (2018). Regional evapotranspiration from image-based implementation of the Surface Temperature Initiated Closure (STIC1.2) model and its validation across an aridity gradient in the conterminous United States, Hydrology and Earth System Sciences, 22, 2311-2341, https://doi.org/10.5194/hess-22-2311-2018. Mallick et al. (2018). Bridging Thermal Infrared Sensing and Physically-Based Evapotranspiration Modeling: From Theoretical Implementation to Validation Across an Aridity Gradient in Australian Ecosystems, Water Resources Research, 54, 3409–3435. https://doi.org/10.1029/2017WR021357. Garcia et al. (2013); https://www.sciencedirect.com/science/article/abs/pii/S0034425712004828.

Morillas et al. (2013); https://agupubs.onlinelibrary.wiley.com/doi/pdf/10.1002/wrcr.20468.

(4) P2: L15-L20 (Introduction). The authors mentioned that LST based ET models

fall into two categories. It is worth mentioning other categories where LST is integrated into Penman-Monteith energy balance (PMEB) equation to directly estimate ET. (5) P3: L12 – L20: I do not see the necessity of such texts. This paper talks about TSEB model improvement and constraining soil evaporation. Yao et al. (2017); Purdy et al. (2018) only used soil moisture data into empirical PT model. I do not see any relevance of these sentences here. The current study is LST based, and the authors should mention why the additional use of SM alongwith LST can produce good ET estimates. (6) P3: L25-L30: This is very significant. Therefore, the texts 'One difficulty lies in developing a consistent representation of the soil evaporation (as constrained by SM, (Chanzy and Bruckler, 1993)), the total ET (as constrained by LST, (Norman et al., 1995)). . ...' should replace the texts in P3 L12 – L20. (7) Site description: Please provide a table describing the characteristics of S1, S2 etc.

(8) Suggesting to provide a Table on main equations of TSEB-SM and sub-equations related to LEsoil; the parameters involved in LEsoil model, their significance, what parameters did you calibrate, what are their value range etc. This would improve the readability of the manuscript.

(9) Section 2.3: There should be a separate sub-section on Retrieval and calibration of rss, arss, brss. The current description is unclear. How the parameters were calibrated? With respect to which observation they were calibrated? All these aspects should be crystallized in the methods section.

(10) There should also be a sub-section on daily ALFApt (Priestley-Taylor parameter) retrieval.

(11) Results and discussion: I am surprised to see the use of old reference (e.g., Sellers et al., 1992). There should be huge amount of literature on soil resistance and soil moisture that deserved citation.

(12) Scatterplot of ALFApt (Priestley-Taylor parameter) versus residual ET and H errors (TSEB-SM – observed) should be shown to reveal the importance of this variable.

[Figure]

(13) How the retrieved soil resistance is related to the residual ET and H errors (TSEB-SM – observed)? What is the magnitude of variability of rss with LST? Such analysis would look excellent.

(14) Residual error analysis should be done to show how the errors in ET and H estimates (TSEB-SM – observed) are related to both DisPATCH soil moisture and observed soil moisture.

I believe the authors put major emphasis to improve the ET and sensible heat flux simulation. But the intermediate parameters should be thoroughly analyzed to give it good scientific quality.

---

## Referee Comment (RC2) · Anonymous Referee #2 · 6 Jun 2019

I read the cited paper, Ait Hssaine et al. AFM 2018b, which is quite similar to this paper. I don't think it is necessary to republished the work on HESS again. Another thing is that TSEB-SM model parameters must be calibrated with ground measurement when they are used to a new region. Both papers do not show how to get ET result at regional scale. Both the abstract has emphasized on calibration of the model parameters. Furthermore, the calibrated parameters are not a fixed value, it varies with time. If the parameters are not fixed value. The model will always need ET or flux measure-

ment to calibrate the parameter. I do not suggest to accept this work for a publication on HESS.

---

## Author Response (AR1)

First, we would like to thank the editor and the two reviewers for the interest they found in our work. Assuring that the comments and suggestions were very helpful for revising and improving our paper. In addition the point by point response to each review, have been taken into account. The 3 main changes in the revised manuscript are summarized below:

1) The writing must be improved as suggested by ref1.

An effort was made to improve clarity and the writing flow and style of the manuscript, notably by 1) re-writing the abstract and some sentences in the Introduction 2) removing several sentences that are out of the context 3) adding separate sub-sections on Retrieval and calibration of $r_{ss}$, $a_{rss}$, $b_{rss}$ and $\alpha_{PT}$

2) The technical details as suggested by ref1 need to be clearly presented and discussed in text.

A sub-section of residual error analysis of the estimated H and LE with the retrieved soil/vegetation parameters was added.

3) The points raised by ref2 are of critical technical concerns that need to be considered carefully. In particular the stability and sensibility of the obtained parameters need to be quantified.

Reviewer 2 was concerned by the model stability. This issue was addressed by varying the initial values $a_{rss,k=0}$ ,and $b_{rss,k=0}$ and by analyzing the sensitivity of the calibration output. In fact, our model converges after a couple of iterations, when the initial values are set to 4.3 and 8.2 for $a_{rss,k=0}$ and $b_{rss,k=0}$, respectively. To further assess the model's stability, the initial values of ($a_{rss,k=0}$, $b_{rss,k=0}$) were randomly set to a range of values between 1 and 13. Such a range (1-13) is based on calibrated values found in the literature (Gentine et al., 2007; Chirouze et al., 2014; Yongjiu and Qingcun, 1997; Griend et al., 1994; Oleson et al., 2008).

Finally, we hope that the above changes satisfied the Editor and the Reviewer's requests.

**Response to Reviewer 1**

The paper titled An evapotranspiration model self-calibrated from remotely sensed surface soil moisture, land surface temperature and vegetation cover fraction: application to disaggregated SMOS and MODIS data' by Ait Hssaine aimed to used LST and disaggregated soil moisture to better constrain the soil evaporation of TSEB model. This is a good idea; however, the presentation of the manuscript needs substantial improvement before being published in HESS. Here are my suggestions and comments, which needs to be considered before being approved for publications.

(1) The abstract is poorly written and does not give a clear message about the novelty of the work. Rework is necessary.

In order to give a clear message about the novelty of the work, an effort was made to re-write the abstract. The revised abstract reads as follows:

Thermal-based two-source energy balance modeling is essential to estimate the land evapotranspiration (ET) in a wide range of spatial and temporal scales. However, the use of thermal-derived land surface temperature ($LST$) is not sufficient to simultaneously constrain both soil and vegetation flux components. Therefore, assumptions (on either soil or vegetation fluxes) are commonly required. To avoid such assumptions, an energy balance model TSEB-SM was recently developed by (?) in order to consider the microwave-derived near-surface soil moisture ($SM$), in addition to the thermal-derived $LST$ and vegetation cover fraction ($f_c$) normally used. While TSEB-SM has been successfully tested using in-situ measurements, this paper represents its first evaluation in real-life using 1 km resolution satellite data, comprised of MODIS (Moderate resolution imaging spectroradiometer) for $LST$ and $f_c$ data and 1 km resolution SM data disaggregated from SMOS (Soil Moisture and Ocean Salinity) observations. The approach is applied during a four-year period (2014-2018) over a rainfed wheat field in the Tensift basin, central Morocco . The field used was seeded for the 2014-2015 (S1), 2016-2017 (S2) and 2017-2018 (S3) agricultural season, while it remained not ploughed (as bare soil) during the 2015-2016 (B1) agricultural season. The classical TSEB model, which is driven only by $LST$ and $f_c$ data, significantly overestimates latent heat fluxes (LE) and underestimates sensible heat fluxes (H) for the four seasons. The overall mean bias values are 119, 94, 128 and 181 W/$m^2$ for LE and -104, -71, -128 and -181 W/$m^2$ for H, for S1, S2, S3 and B1, respectively. Meanwhile, when using TSEB-SM ($SM$ and $LST$ combined data), these errors are significantly reduced, resulting in mean bias values estimated as 39, 4, 7 and 62 W/$m^2$ for LE and -10, 24, 7, and -59 W/$m^2$ for H, for S1, S2, S3 and B1 respectively. Consequently, this finding confirms again the robustness of the TSEB-SM to estimate latent/sensible heat fluxes at large scale by using readily available satellites data. In addition, the TSEB-SM approach has the original feature to allow for calibrating its main parameters (soil resistance and Priestley-Taylor coefficient) from satellite data uniquely, without relying neither on in-situ measurements nor on a priori parameter values.

(2) Introduction: The flow should be logical. Since the objective of the manuscript is to improve the soil evaporation in TSEB to meet up field-scale ET mapping challenges, I do not see any need of line 5 – 10 in page 2.

We agree with the reviewer proposition. The lines 5-10 in page 2 were deleted.

(3) Introduction: 'Evapotranspiration (ET) is a crucial water flux in semi-arid areas'; should be supported by recent literature. The authors should be aware about some recently published ET modeling and mapping studies that particularly addressed the challenges semi-arid and arid ecosystems (that deserves to be cited here); for example, Mallick et al. (2015). Reintroducing radiometric surface temperature into the Penman-Monteith formulation, Water Resources Research, 51, 6214–6243, http://doi.org/10.1002/2014WR016106. Mallick et al. (2014). A surface temperature initiated closure (STIC) for surface energy balance fluxes, Remote Sensing of Environment, 141, 243 - 261. Bhattarai et al. (2019). An automated multi-model evapotranspiration mapping framework using remote sensing and reanalysis data. Remote Sensing of Environment, 229, 69 - 92. Gerhards et al. (2019). Challenges and Future Perspectives of Multi-/Hyperspectral Thermal Remote Sensing for Crop Water Stress Detection: A Review, Remote Sensing, 11(10), 1240; https://doi.org/10.3390/rs11101240. Bhattarai et al (2018). Regional evapotranspiration from image-based implementation of the Surface Temperature Initiated Closure (STIC1.2) model and its validation across an aridity gradient in the conterminous United States, Hydrology and Earth System Sciences, 22, 2311-2341, https://doi.org/10.5194/hess-22-2311- 2018. Mallick et al. (2018). Bridging Thermal Infrared Sensing and Physically-Based Evapotranspiration Modeling: From Theoretical Implementation to Validation Across an Aridity Gradient in Australian Ecosystems, Water Resources Research, 54, 3409–3435. https://doi.org/10.1029/2017WR021357. Garcia et al. (2013); https://www.sciencedirect.com/science/article/abs/pii/S0034425712004828. Morillas et al. (2013); https://agupubs.onlinelibrary.wiley.com

The above recent papers about the modeling and mapping ET are now cited in the revised version. See P2:L2-L3 in the revised manuscript

(4) P2: L15-L20 (Introduction). The authors mentioned that LST based ET models fall into two categories. It is worth mentioning other categories where LST is integrated into Penman-Monteith energy balance (PMEB) equation to directly estimate ET.

To address this issue, the following paragraph was inserted to replace the Lines 15-20 of the revised: In this context, numerous models based on land surface temperature ($LST$) data have been developed such as : (i) residual balance methods that consider ET as the residual term of the energy balance like TSEB (Two-Source Energy Balance, Norman et al., 1995) and SEBS (Surface Energy Balance System, Su et al., 2002), (ii) contextual methods that estimate ET as the potential ET times the evaporative efficiency (Moran et al., 1994) or as the available energy times the evaporative fraction (Merlin et al., 2013; Roerink et al., 2000) and (iii) other categories of models that integrate $LST$ into a water balance model (Olivera et al., 2018) or into Penman-Monteith energy balance (PMEB) equation to directly estimate ET (Amazirh et al., 2017; Mallick et al., 2015)

(5) P3: L12 – L20: I do not see the necessity of such texts. This paper talks about TSEB model improvement and constraining soil evaporation. Yao et al. (2017); Purdy et al. (2018) only used soil moisture data into empirical PT model. I do not see any relevance of these sentences here. The current study is LST based, and the authors should mention why the additional use of SM along with LST can produce good ET estimates.

We agree with the reviewer's opinion, and accordingly the texts on page 3 Lines 12-30 were removed.

(6) P3: L25-L30: This is very significant. Therefore, the texts 'One difficulty lies in developing a consistent representation of the soil evaporation (as constrained by SM, (Chanzy and Bruckler, 1993)), the total ET (as constrained by LST, (Norman et al., 1995)): : :..' should replace the texts in P3 L12 – L20.

Agreed and done.

(7) Site description: Please provide a table describing the characteristics of S1, S2 etc.

According to the Reviewer's suggestion a table describing the characteristics of the 4 agricultural seasons was presented (Table 1 in the revised manuscript).

**Table 1.** Characteristics of the study site.

| Study period | R ainfall amount (mm) | Field status |
|---|---|---|
| Oct 2014- Jun 2015 (S1) | 608 | Cultivated |
| Aug 2015- Sep 2016 (B1) | 157 | Bare soil |
| Sep 2016- Jun 2017 (S2) | 214 | Cultivated |
| Oct 2017- Jun 2018 (S3) | 481 | Cultivated |

(8) Suggesting to provide a Table on main equations of TSEB-SM and sub-equations related to LEsoil; the parameters involved in LEsoil model, their significance, what parameters did you calibrate, what are their value range etc. This would improve the readability of the manuscript.

According to the Reviewer's suggestion, a new Table (Table 2) was inserted to present the main equations of TSEB-SM and the sub-equations related to LEsoil and LEveg.

**Table 2.** Mean equations of TSEB-SM.

| Variable | E quation | Value range |
|---|---|---|
| Soil heat flux | $LE_{soil} = \dfrac{\rho c_p}{\gamma} \dfrac{e_s - e_a}{r_{ah} + r_s + r_{ss}}$ | 0-600 W/$m^2$ |
| Resistance to vapor diffusion in the soil | $r_{ss} = \exp(a_{rss} - b_{rss} \times \dfrac{SM}{SM_{sat}}$ | $a_{rss}$ and $b_{rss}$ : (1-13) |
| Soil moisture at saturation | $SM_{sat} = 0.1 \times (-108 \times f_{sand} + 49.305)$ | 0,47$m^3/m^3$ |
| Cost function for minimizing $r_{ss}$ | $F_{inst} = (T_{surf,sim} - T_{surf,mes})$ | $F_{inst} = 5$ K |
| Vegetation latent heat flux | $\alpha_{PT}.f_g.\dfrac{\triangle}{\triangle + \gamma}.R_{n,veg}$ | $\alpha_{PT}$ (0-2) $f_g$ =1 |

(9) Section 2.3: There should be a separate sub-section on Retrieval and calibration of $r_{ss}$, $a_{rss}$, $b_{rss}$. The current description is unclear. How the parameters were calibrated? With respect to which observation they were calibrated? All these aspects should be crystallized in the methods section.

A detailed description and the main equations used for the calibration of the two soil parameters $a_{rss}$ and $b_{rss}$ as well as $\alpha_{PT}$ have been presented in our article published in AFM 2018 (Ait Hssaine et al., 2018b). For clarity, the lines 14-18 in page 9 have been restructured as follows:

'In Ait Hssaine et al. (2018b), an innovative calibration approach of $\alpha_{PT}$, $a_{rss}$ and $b_{rss}$ is developed from in-situ SM, LST and $f_c$ data (Ait Hssaine et al., 2018b). The calibration methodology is briefly reminded below.

2.3.1.1. Retrieval and calibration of $r_{ss}$, $a_{rss}$ and $b_{rss}$

The $r_{ss}$ is first adjusted by minimizing a cost function defined by:

$$F_{inst} = (T_{surf,sim} - T_{surf,mes})^2 \tag{1}$$

With $T_{surf,sim}$ and $T_{surf,mes}$ being the simulated and measured LST, respectively. The inverted $r_{ss}$ is then correlated to the SM (in-situ or DisPATCh) to determine the $a_{rss}$ and $b_{rss}$ parameters by considering that, when $f_c$ is lower than a given threshold ($f_{c,thres}$), the dynamics of total LE is mainly controlled by the temporal variation of soil evaporation. Meaning that both soil parameters are estimated when the PT coefficient can be set to a constant value.'

(10) There should also be a sub-section on daily ALFApt (Priestley-Taylor parameter) retrieval.

The P7: L18-L21 was replaced by a sub-section

2.3.1.2. Daily $\alpha_{PT}$ retrieval

Once the soil resistance has been calibrated, the PT coefficient is retrieved on a daily basis when $f_c$ is larger than $f_{c,thres}$, by minimizing a cost function at the Terra and Aqua-MODIS overpass times:

$$F_{daily} = \sum (T_{surf,sim} - T_{surf,mes})^2 \tag{2}$$

In fact, an iterative loop is run on soil ($r_{ss}$) and vegetation ($\alpha_{PT}$) parameters to reach convergence of all parameters.

(11) Results and discussion: I am surprised to see the use of old reference (e.g., Sellers et al., 1992). There should be huge amount of literature on soil resistance and soil moisture that deserved citation.

According to Reviewer's suggestion, the following papers were cited in the revised version (P7:L4): Li et al., 2006; Chirouze et al., 2014

(12) Scatterplot of ALFApt (Priestley-Taylor parameter) versus residual ET and H errors (TSEB-SM – observed) should be shown to reveal the importance of this variable.

Figure 1 (Figure 12 in the revised manuscript) plots retrieved $\alpha_{PT}$ vs residual H and LE error. The retrieved $\alpha_{PT}$ is poorly correlated to residual H (R=-0.27) and ET (R=0.27) errors especially for the seasons S1 and S2. For the season S3, few retrieved $\alpha_{PT}$ values were available because of the non-availability of MODIS products during cloudy days. It is shown that the trend between $\alpha_{PT}$ and residual H error is slightly negative for S1 while it is slightly positive for S2. According to these results, no information linked to the variability of $\alpha_{PT}$ versus residual ET and H errors can be derived.

(13) How the retrieved soil resistance is related to the residual ET and H errors (TSEB-SM – observed)? What is the magnitude of variability of $r_{ss}$ with LST? Such analysis would look excellent.

Figure 2 (Figure 13 in the revised manuscript) plots retrieved $r_{ss}$ vs residual H and LE error and $LST$ for the four study periods. The retrieved $r_{ss}$ is negatively correlated (R=-0.33) with residual H error (predicted–observed) for the four seasons, while it is positively correlated (R=0.33) with residual LE error. The residual error covers a wide range (between -150 and 150 $W/m^2$) for the lower $r_{ss}$ values, while it is biased for the higher $r_{ss}$ values. Such a result indicates that the $r_{ss}$ formulation as a function of near-surface SM needs further improvements (Merlin et al., 2016;2018) in order to reduce systematic uncertainties in evaporation estimates, especially in dry (moisture-limited) conditions. Consistent with the general decrease of $LST$ with $SM$, $LST$ is positively correlated to retrieved $r_{ss}$ (R=0.45). This is very coherent since $r_{ss}$ decreases with the increase of $SM$.

(14) Residual error analysis should be done to show how the errors in ET and H estimates (TSEB-SM – observed) are related to both DisPATCH soil moisture and observed soil moisture.

Regarding the sensitivity analysis of residual H and LE errors to observed $SM$, Figure 3 (Figure 14 in the revised manuscript) shows that $SM$ is positively correlated with residual H error, while it is negatively correlated with residual LE error for the

entire study period. The correlation coefficient is about 0.3 when using DisPATCh $SM$ while it is about 0.4 when using in-situ $SM$. This difference can be explained by the uncertainty (including spatial representativeness issues at the localized scale) in DisPATCh $SM$. The positive correlation coefficient between residual LE error and observed $SM$ is likely to be due to the systematic errors in $r_{ss}$ estimates for dry conditions as mentioned previously. For S2, B1 and S3, the residual H error ranges

5    between -150 and 50 W/$m^2$ for SM between 0-0.10 $m^3/m^3$ while it is slightly overestimated for the higher range of $SM$. The residual LE error is also found to be influenced by $SM$, but in the opposite sense.

I believe the authors put major emphasis to improve the ET and sensible heat flux simulation. But the intermediate parameters should be thoroughly analyzed to give it good scientific quality.

[Figure]

**Figure 1.** $\alpha_{PT}$ vs Residual H and LE error .

[Figure]

**Figure 2.** $r_{ss}$ vs Residual H and LE error and LST for the four study periods .

[Figure]

**Figure 3.** Residual H and LE error vs SM .

**Response to Reviewer 2**

I read the cited paper, Ait Hssaine et al. AFM 2018b, which is quite similar to this paper. I don't think it is necessary to republished the work on HESS again.

We emphasize that our previous paper published in AFM 2018 was a feasibility study of the TSEB-SM algorithm using field measurements. In this article, we tested the same algorithm in real life using readily available satellite thermal and microwave data. Such an application is fully original and raises new research questions clearly not addressed elsewhere. To our knowledge, there is still no evapotranspiration model that has been coupled to remotely sensed soil moisture and land surface temperature at such high (1 km) resolution. This is now possible using the formalism of TSEB-SM (Ait Hssaine et al. 2018) and the high-resolution soil moisture data sets derived from the disaggregation of SMOS-like data (Molero et al. 2016, Peng et al. 2017). To clarify this point, the following sentence was inserted as soon as the abstact of the revised : "While TSEB-SM has been successfully tested using in-situ measurements, this paper represents its first evaluation in real-life using 1 km resolution satellite data, comprised of MODIS (Moderate resolution imaging spectroradiometer) for $LST$ and $f_c$ data and 1 km resolution SM data disaggregated from SMOS (Soil Moisture and Ocean Salinity) observations".

Another thing is that TSEB-SM model parameters must be calibrated with ground measurement when they are used to a new region.

We totally disagree with the Reviewer. The main feature of the TSEB-SM approach actually is to calibrate two fundamental parameters (controlling the soil evaporation and the plant transpiration separately) from soil moisture, land surface temperature and vegetation cover fraction data. In this study, all the data used for calibration are derived from remote sensing. Therefore, we do not rely on any in situ measurement, as notably stated by the title.

To address this concern, we clarified as soon as the abstract and introduction, the remote sensing-based nature of the TSEB-SM approach.

In the previous manuscript (P4: L13-L14): "The objective of this study is to investigate how satellite data can be used to retrieve the main parameters ($a_{rss}$, $b_{rss}$ and $\alpha_{PT}$) of TSEB-SM model." Was replaced (in the new manuscript, P4: L4-L6) by: "TSEB-SM has the cutting edge capability to calibrate its main parameters from remotely sensed data, but the real-life application has not yet been tested. The objective of this paper is thus to demonstrate for the first time this capacity using disaggregated SMOS and MODIS (Moderate resolution imaging spectroradiometer) data".

Both papers do not show how to get ET result at regional scale.

It is true that the paper Ait Hssaine et al. (2018) does not test the TSEB-SM approach using remote sensing data. This is precisely why the present study is essential to demonstrate the capacity of the model to provide ET results at regional scale. Below, we present an example of a map of evapotranspiration in our area of study 'El Haouz' derived from the same model TSEB-SM

[Figure]

**Figure 4.** Map of Evapotranspiration using MODIS LST and DisPATCh SM .

The spatial application of ET will not be introduced in the manuscript. Since, this approach will be used in another ongoing study with finer resolution, compatible with the land use of the agricultural zone around Marrakech.

Both the abstract has emphasized on calibration of the model parameters. Furthermore, the calibrated parameters are not a fixed value, it varies with time. If the parameters are not fixed value. The model will always need ET or flux measurement to 5 calibrate the parameter. I do not suggest to accept this work for a publication on HESS.

It is true that $\alpha_{PT}$ varies in time. However, this parameter is estimated from remote sensing data. The flux measurements are needed only for the validation of the TSEB-SM simulated sensible and latent heat fluxes. To clarify this point, the following sentences are inserted as soon as the introduction (P3: L28-L30): "It should be noted that only LST and SM are used for the calibration of yearly $a_{rss}$ and $b_{rss}$ as well as daily $\alpha_{PT}$, while the flux measurements are needed only for the validation of the 10 TSEB-SM simulated sensible and latent heat fluxes."

We hope that the above changes have clarified the self-calibrated TSEB-SM approach and its implementation using readily available remote sensing data.

---

## Author Response (AR2)

**Response to Editor**

Dear authors, Thank you very much for submitting your revision. In addition to the two reviews, there was a third one that also needs to be addressed. Please provide a point by point response and your revision thereafter.

I briefly checked some of your results and was puzzled by the ranges of values in $r_{ss}$. In Fig. 5, $\log(r_{ss})$ is 0~15, while in Fig. 13, $r_{ss}$ is (0 ~2)x10$^4$, or $\log(r_{ss})$~4. Please clarify these issues.

Thank you for pointing this out. In our case, we used the natural logarithm and not the decimal logarithm. For $r_{ss}$ (0~2)x10$^4$, the variation of $\ln(r_{ss})$ is hence 0~9.9. For clarity, "log" is systematically replaced by "ln" in the revised manuscript. Consistently, the values presented in the revised Fig. 13 (see below) correspond to the values obtained in figure 5.b. Note that few values are missing in Fig. 13 because of gaps in measured H/LE.

Another issue is related to the soil heat flux. Please add a sentence to explain how it was estimated.

15

The equation used to estimate the soil heat flux was added to the revised manuscript (Page 6 Lines 25-27): "The surface soil heat flux is estimated as a fraction of $R_{n,soil}$:

$$G = c_g * R_{n,soil} \tag{1}$$

where cg ~ 0.35" (Choudhury et al., 1987)

Please also pay attention to table 2, it seems the soil heat flux is the latent heat flux of soil surface (or soil latent heat).

The soil heat flux in Table 2 is changed by Soil latent heat flux

25

The eq. for $r_{ss}$ is not complete.

Checked and corrected

[Figure]

**Figure 1.** Previous version of fig.13 .

[Figure]

**Figure 2.** Revised fig.13 .

**Response to Reviewer 1**

Parameters in TSEB-SM were calibrated by using MODIS LST and SMOS SM observations. This gives us a solution on how to combine the use LST and SM for the ET simulation. This work is innovative and has a potential to be used by other studies. As an open access publication journal, I suggest the authors open the access to the model code and input data used in the paper, or published them with the paper on HESS.

First, we thank the reviewer for his/her interest in our work. Regarding the access to the model code, we would like to mention that the current version is in Matlab. We are planning to translate it into Python before the source code can be released under a license in which the copyright holder grants users the rights to study, change, and distribute the software to anyone and for any purpose. However, this step will be considered separately from the publication process of the HESS paper.

There are some minor mistakes in the manuscript, which are listed below:
In the same way, Li et al. (2006) indicated that the model performance is sensitive to these two coefficients, which two coefficients? SM and LST? Why SM and LST are called coefficients?

These two coefficients are $a_{rss}$ and $b_{rss}$. To clarify this point, the following sentence (lines 5-6 page 3 of the previous version): "Moreover, the soil evaporation is constrained by the SM through soil-texture dependent coefficients reported in (Sellers et al., 1992). In the same way, Li et al. (2006) indicated that the model performance is sensitive to these two coefficients".

Was replaced by (Page 3 Line 4-6 of the revised): "Moreover, the soil evaporation is constrained by the SM through its soil-texture dependent coefficients ($a_{rss}$ and $b_{rss}$) reported in (Sellers et al., 1992). In the same way, Li et al. (2006) indicated that the model performance is sensitive to these two coefficients".

Rewrite the sentence "TSEB-SM is applied to 1 km resolution using MODIS LST/ fc data and to SMOS SM data is applied."

The sentence was modified (Page 4 Line 7 of the revised) : " TSEB-SM is applied at 1 km resolution using MODIS LST, MODIS fc and disaggregated SMOS SM data".

Eq.7. When LST was simulated, what kind of input data were used and which values for the $\alpha_{PT}$, $r_{ah}$, $r_s$ were given? This information is important when you take LST as a simulation output. It's better to include this information.

For clarity, the details regarding the LST estimation were inserted in Page 7 Lines 21-27:
"The LST (noted $T_{surf,sim}$)) was simulated as follows:

$$(T_{surf,sim} = (f_c * (T_{veg})^4 + (1 - f_c) * (T_{soil})^4)^{0.25} \tag{2}$$

where $T_{veg}$ and $T_{soil}$ are the vegetation and soil components of temperature (K). The LST is simulated each 30 min (between 11 am and 2 pm) and at Terra and Aqua overpass times for in-situ and satellite data, respectively.
The LST at first calibration step is simulated with a constant value of $\alpha_{PT}$ (average value of the $\alpha_{PT}$ retrieved for $f_c > 0.5$). Then, for the second calibration step, it is simulated using the daily retrieved $\alpha_{PT}$."
The details regarding the resistances were inserted in Page 7 Lines 4-8:
" $r_{ah}$ the aerodynamic resistance is calculated from the adiabatically corrected logarithmic temperature profile equation (Brutsaert, 1982) and $r_s$ the surface-soil resistance to transport of heat between the soil surface and a height representing the canopy, is estimated using (Sauer et al., 1995). Both resistances are simulated each 30 min (between 11 am and 2 pm) and at Terra and Aqua overpass times for in-situ and satellite data, respectively." A detailed description of TSEB model and the main equations used for the calibration strategy have been presented in our article published in AFM 2018 (Ait Hssaine et al., 2018)."

Do you mean rss or $a_{rss}$, $b_{rss}$ with "the soil resistance" by "Once the soil resistance has been calibrated"?

The soil resistance refers to $r_{ss}$. To clarify this point, the following explanation was inserted in the revised (Page 8 Line 2): "once both parameters $a_{rss}$ and $b_{rss}$ have been estimated"

What is the difference of both $T_{surf,mes}$ in eq.7 and 8? Both Terra and Aqua LST were used in eq. 8. How about eq. 7? Which LST were used in eq. 7? Or both were used in eq.7?

$T_{surf,mes}$ in eq.7 are instantaneous LST each 30 min (between 11 am and 2 pm) while using in-situ data and at Terra and Aqua overpass times when using satellite data. $T_{surf,mes}$ in eq.8 is daily LST (an average of such LST measurements for a given day).

There are some mistakes in table2. It is not Soil heat flux.

Thank you for pointing out this error. The error was corrected.

It might be interesting to show the time series of soil and canopy H, LE component against flux measurement to look at the influence of dynamic variation with the variational $\alpha_{PT}$ ,and $a_{rss}$, $b_{rss}$.

We fully agree with the Reviewer. The time series of soil and canopy H, LE component against flux measurement is of great interest for our work. However, our focus for this work is the validation of total H and LE. As we mentioned (Response to Reviewer 3), measurement devices of separate E/T were not present at the Sidi Rahal site. Note that an intensive experiment was recently undertaken in the same region (Rafi et al. 2019) to provide such E/T estimates over irrigated wheat crops. Our plan is to develop an approach to implement TSEB-SM at the 100 m resolution and to validate the E/T components over those fields. This will be the focus of future studies.

We point out that there is no dynamic variation of $a_{rss}$ and $b_{rss}$, as we only retrieve a pair of values calculated for the entire study period.

To look at the influence of dynamic variation with the variational $\alpha_{PT}$. We plot the time series of simulated and measured LE superimposed with daily retrieved $\alpha_{PT}$. At the beginning of the season, the soil is almost bare meaning that the $\alpha_{PT}$ doesn't affect the LE variabilities. In the growing stage, both observed and simulated fluxes depict a similar behaviour. The maximum of $LE_{sim}$ is reached after the peak of rainfall during the growing stage which coincides exactly with the maximum value of the retrieved $\alpha_{PT}$. Afterward, $LE_{sim}$ tends to decrease at the senescence stage because of the decrease in daily $\alpha_{PT}$

[Figure]

**Figure 3.** Time series of simulated and measured LE superimposed with daily retrieved $\alpha_{PT}$ .

**Response to Reviewer 2**

The revised paper titled 'An evapotranspiration model self-calibrated from remotely sensed surface soil moisture, land sur-
face temperature and vegetation cover fraction: application to disaggregated SMOS and MODIS data' by Ait Hssaine have
substantially improved as compared to the initial version. However, couple of points need to be addressed after which it can be
accepted for publication.

Introduction: The first paragraph should be written as follows (some references need to be added): Evapotranspiration (ET)
is a crucial water flux for drought monitoring (Bhattarai et al., 2019; Gerhards et al., 2019; Mallick et al., 2014, 2016, 2018),
water resource management (Please add references related to METRIC model applications in water resources management)
and climate simulation (Littell et al., 2016; Molden et al., 2010) in the semi-arid ecosystems. A precise estimate of ET deter-
mines the crop water requirements, which subsequently allows the optimization of irrigation water applications (Allen et al.,
1998). Reference: Mallick et al. (2018). Bridging Thermal Infrared Sensing and Physically-Based Evapotranspiration Model-
ing: From Theoretical Implementation to Validation Across an Aridity Gradient in Australian Ecosystems, Water Resources
Research, 54, 3409–3435. https://doi.org/10.1029/2017WR021357.

We thank the reviewer for the precise review of our paper. The paragraph in the previous manuscript was changed accord-
ing to the above suggestions.

Page 2 L13-14 (Introduction): Please correct it as, (iii) other categories of models that integrate LST into water balance models
(Olivera-Guerra et al., 2018) or into the Penman-Monteith Energy Balance (PMEB) equation to directly estimate ET (Amazirh
et al., 2017; Mallick et al., 2015, 2018).

The sentence in Page 2 L13-14 (Introduction) was corrected.

Page 2 L24 – 28 (Introduction): Please correct as follows: Recently, Boulet et al. (2015) have developed the Soil-Plant-
Atmosphere and Remote Sensing Evapotranspiration (SPARSE) model, which is similar to the TSEB model in its basic as-
sumption, but, with additional constraints to improve the ET model performance in heterogeneous vegetation.

Lines L24-28 in page 2 (Introduction) were corrected in the revised manuscript.

P4 L5 – 10: Calling it cutting edge capability is an overstatement. Please correct it as follows:
Although TSEB-SM has the capability to calibrate its main parameters from remotely sensed data, however, the real-life
application needs extensive evaluation and testing. The objective of this paper is thus to demonstrate for the first time this
capacity using disaggregated SMOS and MODIS (Moderate resolution imaging spectroradiometer) data.

Line 5-10 in page 4 were corrected in the revised version.

Conclusions: Some insightful conclusions are expected. Better to give the significant conclusions in bullet points The con-
clusion is rewritten in bullet point as suggested by the Reviewer.

The microwave-derived near-surface soil moisture (SM) from SMOS and the thermal-derived land surface temperature (LST)
from MODIS are integrated simultaneously in the TSEB formalism within a calibration procedure to invert both the soil resis-
tance to evaporation (constant parameters) and the PT coefficient based on a threshold on $f_c$. The TSEB-SM model is applied
during a four-year period (2014-2018) over a rainfed wheat field in the Tensift basin, central Morocco. Significant conclusions
are given below.

The constraint applied on the soil evaporation represented explicitly as a function of SM via a soil resistance term reduces
the errors when using TSEB-SM instead of TSEB.

-The first step of the TSEB-SM approach is to calibrate $r_{ss}$ for ($f_c \leq f_{c,thres}$) at Terra and Aqua overpass times. Despite the scale difference between the MODIS/DisPATCh resolution data and the footprint size of in-situ measurements, the parameters ($a_{rss}$, $b_{rss}$) calculated for the entire study period using satellite data are relatively close to those derived from in-situ measurements.

-The second step of the TSEB-SM approach is to invert the $\alpha_{PT}$ on a daily basis for $f_c > f_{c,thres}$ by using LST and SM data. The maximum of daily calibrated $\alpha_{PT}$ are 1.38, 1.25 and 0.87, when using satellite data, for S1, S2 and S3, respectively. Those values are in accordance with the total rainfall amounts, which were about 608, 214 and 421mm/wheat season for S1, S2 and S3 respectively. S1 and S2 have the same distribution of daily calibrated $\alpha_{PT}$ when comparing with the $\alpha_{PT}$ retrieved using in-situ data, while the retrieved $\alpha_{PT}$ remains at a mostly constant value ( 0.7) throughout the study period S3 because of the non-availability of MODIS products during cloudy days.

-Finaly, an analysis of the spatial distributions and the magnitude of the turbulent fluxes using remotely sensing data produced from the two models were conducted. TSEB exhibits larger errors on H and LE estimates. These uncertainties can be linked to the theoretical value of $\alpha_{PT}$, which is fixed to 1.26 for the whole study period, as well as to the scale mismatch between the 1 km resolution of MODIS LST and the footprint size (approximately 1 m) of the ground-based radiometer.

As a short term prospect, the use of high-resolution products from active sensors (Sentinel-1) would allow applying the TSEB-SM approach at the field scale over heterogeneous (e.g. irrigated) landscapes. Also the robustness of TSEB-SM in terms of evaporation/transpiration partitioning will be tested by using independent flux measurements derived from lysimeters and sap flow sensors (Rafi et al., 2019). In addition, the evaluation of ET at large scale is missing. Spatialized measurements that could be collected by scintillometers installed at various points in the region would be a solution for that purpose.

**Response to Reviewer 3**

Two-source energy balance (TSEB) model based on remotely sensed land surface temperature is an important modelling approach for estimating evapotranspiration (ET) and its components of E and T. This study aimed to use disaggregated soil moisture from SMOS to constrain the soil evaporation in TSEB and to further improve ET estimation. I do agree this is a good start point. However, the disaggregated soil moisture data should be reliable to calibrate the TSEB-SM model. If the authors failed to prove the reliability of the disaggregated soil moisture, this manuscript is just like the Ait Hssaine et al. AFM 2018b and should not be published again. After carefully read this manuscript, I think the "proof" provided in the current manuscript is not strong enough.

It is true that the accuracy in the disaggregated soil moisture data fosters the possibility to efficiently constrain the soil evaporation in TSEB. However, we do not agree with the reviewer that the accuracy in DisPATCh data is not sufficient to show the strength of the approach. In this paper, we show that the use of DisPATCh soil moisture improves the TSEB results (See results in Tables 4 and 5 and Figures 8, 9, 10 and 11). In addition, the DisPATCh products have been extensively evaluated, especially over semi-arid areas like the Marrakech region (Bandara et al., 2015; Colliander et al., 2017; Escorihuela et al., 2018; Escorihuela and Quintana-Seguí, 2016; Lievens et al., 2015; Malbéteau et al., 2016, 2018; Merlin et al., 2012, 2013, 2015; Molero et al., 2016; Ojha et al., 2019; Peng et al., 2017; Sabaghy et al., 2020). In our opinion, both arguments 1) improved TSEB results by including DisPATCh soil moisture and 2) extensive validation of DisPATCh data over semi-arid areas) are sufficiently strong to prove that the approach using DisPATCh is reliable.

(1) The DisPATCh algorithm was applied to disaggregated 1km resolution soil moisture from SMOS soil moisture products. The DisPATCh algorithm is actually based on a contextual method to estimate soil and vegetation evaporative fraction. The question is whether it is reliable to apply the contextual method to such as heterogeneous farmland (See Figures 1 and 4)?

The contextual method in DisPATCh is implemented at 1 km resolution within a SMOS pixel i.e. within a 40 km by 40 km area (Merlin et al., 2012). Over the study area, the heterogeneity of the surface (in terms of vegetation cover and soil moisture conditions) is large due to the presence of both dry land and irrigated areas near Marrakech. In such heterogeneous conditions, the contextual approach is very well adapted as the dry and wet boundaries can be well estimated all along the agricultural season.

It seems the authors only use one soil moisture site to do the validation. Although the authors provide a scatter in Figure 3, this is not sufficient.

We fully agree with the reviewer that one single soil moisture site is not sufficient to assess the accuracy of the DisPATCh products. Instead our choice to use the DisPATCh product is supported by the numerous evaluation studies of this product (Bandara et al., 2015; Colliander et al., 2017; Djamai et al., 2015; Escorihuela et al., 2018; Escorihuela and Quintana-Seguí, 2016; Lievens et al., 2015; Malbéteau et al., 2016, 2018; Merlin et al., 2012, 2013, 2015; Molero et al., 2016; Ojha et al., 2019; Peng et al., 2017; Sabaghy et al., 2020) . Actually the scatterplots of Figure 3 aim to simply verify that the DisPATCh soil moisture is consistent, during four agricultural seasons, at the site level where the comparison between TSEB and TSEB-SM is undertaken.

To further emphasize this point, the following sentence was inserted (Page 9 Lines 14-19 of the revised) : " The DisPATCh products have been extensively evaluated, especially over semi-arid areas like the Marrakech region (Bandara et al., 2015; Colliander et al., 2017; Djamai et al., 2015; Escorihuela et al., 2018; Escorihuela and Quintana-Seguí, 2016; Lievens et al., 2015; Malbéteau et al., 2016, 2018; Merlin et al., 2012, 2013, 2015; Molero et al., 2016; Ojha et al., 2019; Peng et al., 2017; Sabaghy et al., 2020). Actually the scatterplots of Figure 3 aim to verify that the DisPATCh soil moisture is consistent, during four agricultural seasons, at the site level where the comparison between TSEB and TSEB-SM is undertaken".

(2) The manuscript failed to provide related information to the DisPATCh algorithm when applying the algorithm to the study

area. How the dry and wet edge associated with the DisPATCh algorithm been determined in this study?

The description of the DisPATCh algorithm is out of the scope of this paper. Instead we are using the DisPATCh product as input to TSEB formalism. As a short reminder of the edge determination in DisPATCh, the following information were inserted in the revised manuscript (Page 6 Lines 4-7): "Soil ($T_{s,min}$, $T_{s,max}$) and vegetation ($T_{v,min}$, $T_{v,max}$) temperature endmembers are estimated from the polygon obtained by plotting MODIS LST against MODIS NDVI, where the LST is partitioned into its soil and vegetation components according to the trapezoid method of Moran et al. (1994) (Moran et al., 1994). Details on the DisPATCh algorithm and the methodology to determine dry and wet edges can be found in Merlin et al. (2012). The aim of this work is to use the disaggregated SM to feed our model TSEB-SM."

The EFs derived from empirical contextual based DisPATCh algorithm is applied to disaggregated SMOS soil moisture, and the soil moisture is applied to constraint a more physically based TSEB ET model. What is the difference between EFs derived from DisPATCh and TSEB? Which model is more reliable?

The evaporative fractions derived from DisPATCh and from TSEB are not the same quantity. DisPATCh is based on the soil evaporative efficiency (SEE, ratio of actual to potential soil evaporation), which is used as a proxy for the 0–5 cm (microwave-derived) soil moisture variability. While in TSEB-SM we use the surface evaporative fraction (EF, defined as the ratio of latent heat to available energy) to normalize the output fluxes using the LST-derived available energy.

EF and SEE are both indicators of SM. However, the comparison of EF and SEE proxies indicates that SEE is more directly related to SM, especially for the higher range of SM values. The diurnal variability of EF, due to variations in incident radiation and relative humidity, seems to explain the superiority of the SEE-based approach for SM disaggregation purposes compared to EF-based approach (Merlin et al., 2008). For clarity, we plot the EF versus in-situ SM and the DisPATCh SM (linked to SEE) versus in-situ SM (The figure below). The figure shows the good agreement between DisPATCh SM and in-situ SM ($R^2$=0.59) compared to EF versus in-situ SM ($R^2$=0.31). EF is more scattered for higher SM values.

[Figure]

**Figure 4.** DiSPATCh SM (linked to SEE) versus in situ SM and TSEB EF versus in situ SM .

(3) It is clear the study area is very small from the Map scale in Figure 1, and the MODIS products with coarse resolution may not be a good choice in this area. I should recommend the authors try to use other high-resolution satellite images.

We agree with the reviewer that the study area is small compared to the 1 km resolution MODIS pixel. However, our choice was not arbitrary, because the field is located within a larger area occupied, quite uniformly, by rainfed wheat 'Bour'. This field was chosen to be representative at a scale larger than 1 km, thus enabling the comparison between 1 km resolution satellite-derived and localized in-situ measurements. The use of high-resolution (Landsat) satellite images, compatible with the land use of the irrigated agricultural zone around Marrakech will be the focus of another separate paper. Note that another SM product should be used in this case (e.g. Ojha et al. 2019).

(4) As a two-source model, TSEB can provide ET components (E and T) as well. The authors only provide validation results for H and LE. It should be very interesting if the E/T partition can be provided for the original TSEB and the calibrated one.

We fully agree with the Reviewer. The validation of the E/T partition estimated by the original TSEB and by the calibrated one (TSEB-SM) is of great interest for our work. However, measurement devices of separate E/T were not present at the Sidi Rahal site. Note that an intensive experiment was recently undertaken in the same region (Rafi et al. 2019) to provide such E/T estimates over irrigated wheat crops. Our plan is to develop an approach to implement TSEB-SM at the 100 m resolution and to validate the E/T components over those fields and compare the results of partition with those of original TSEB. This will be the focus of future studies.

(6) More consideration should be taken to the smoothed PT coefficient in Figure 6. It seems the points are quite scatter.

It is true that the retrieved PT coefficients are quite scattered. A particular care was given to reduce uncertainties. To remove outliers and to reduce uncertainties on the daily inverted $\alpha_{PT}$, these values were smoothed by using weighted linear least squares and a 1st degree polynomial model while taking a 30-day sliding average over the entire season.

Other specific comments (1) P6 Line 6, "the revised soil temperature" –may be retrieved?

Corrected.

(2) Table 2. The variable for the first equation should be soil latent heat flux. Soil heat flux should be G

Done.

**References**

Ait Hssaine, B., Merlin, O., Rafi, Z., Ezzahar, J., Jarlan, L., Khabba, S., and Er-Raki, S.: Calibrating an evapotranspiration model using radiometric surface temperature, vegetation cover fraction and near-surface soil moisture data, Agricultural and Forest Meteorology, 256-257, 104–115, https://doi.org/10.1016/j.agrformet.2018.02.033, 2018.

5 Bandara, R., Walker, J. P., Rüdiger, C., and Merlin, O.: Towards soil property retrieval from space: An application with disaggregated satellite observations, Journal of Hydrology, 522, 582–593, https://doi.org/10.1016/j.jhydrol.2015.01.018, 2015.

Brutsaert, W.: Introduction, in: Evaporation into the Atmosphere, pp. 1–11, Springer Netherlands, Dordrecht, https://doi.org/10.1007/978-94-017-1497-6_1, http://link.springer.com/10.1007/978-94-017-1497-6{_}1, 1982.

Choudhury, B., Idso, S., and Reginato, R.: Analysis of an empirical model for soil heat flux under a growing wheat crop for esti-
10 mating evaporation by an infrared-temperature based energy balance equation, Agricultural and Forest Meteorology, 39, 283–297, https://doi.org/10.1016/0168-1923(87)90021-9, http://linkinghub.elsevier.com/retrieve/pii/0168192387900219, 1987.

Colliander, A., Fisher, J. B., Halverson, G., Merlin, O., Misra, S., Bindlish, R., Jackson, T. J., and Yueh, S.: Spatial Downscaling of SMAP Soil Moisture Using MODIS Land Surface Temperature and NDVI during SMAPVEX15, IEEE Geoscience and Remote Sensing Letters, 14, 2107–2111, https://doi.org/10.1109/LGRS.2017.2753203, 2017.

15 Djamai, N., Magagi, R., Goita, K., Merlin, O., Kerr, Y., and Walker, A.: Disaggregation of SMOS soil moisture over the Canadian Prairies, Remote Sensing of Environment, 170, 255–268, https://doi.org/10.1016/j.rse.2015.09.013, 2015.

Escorihuela, M. J. and Quintana-Seguí, P.: Comparison of remote sensing and simulated soil moisture datasets in Mediterranean landscapes, Remote Sensing of Environment, 180, 99–114, https://doi.org/10.1016/j.rse.2016.02.046, 2016.

Escorihuela, M. J., Merlin, O., Stefan, V., Moyano, G., Eweys, O. A., Zribi, M., Kamara, S., Benahi, A. S., Abdallahi, M., Ebbe, B.,
20 Chihrane, J., Ghaout, S., Cissé, S., Diakité, F., Lazar, M., Pellarin, T., Grippa, M., Cressman, K., and Piou, C.: SMOS based high resolution soil moisture estimates for desert locust preventive management, Remote Sensing Applications: Society and Environment, 11, 140–150, https://doi.org/10.1016/j.rsase.2018.06.002, https://doi.org/10.1016/j.rsase.2018.06.002, 2018.

Li, F., Kustas, W. P., Anderson, M. C., Jackson, T. J., Bindlish, R., and Prueger, J. H.: Comparing the utility of microwave and thermal remote-sensing constraints in two-source energy balance modeling over an agricultural landscape, Remote Sensing of Environment, 101,
25 315–328, https://doi.org/10.1016/j.rse.2006.01.001, 2006.

Lievens, H., Tomer, S. K., Al Bitar, A., De Lannoy, G. J., Drusch, M., Dumedah, G., Hendricks Franssen, H. J., Kerr, Y. H., Martens, B., Pan, M., Roundy, J. K., Vereecken, H., Walker, J. P., Wood, E. F., Verhoest, N. E., and Pauwels, V. R.: SMOS soil moisture assimilation for improved hydrologic simulation in the Murray Darling Basin, Australia, Remote Sensing of Environment, 168, 146–162, https://doi.org/10.1016/j.rse.2015.06.025, 2015.

30 Malbéteau, Y., Merlin, O., Molero, B., Rüdiger, C., and Bacon, S.: DisPATCh as a tool to evaluate coarse-scale remotely sensed soil moisture using localized in situ measurements: Application to SMOS and AMSR-E data in Southeastern Australia, International Journal of Applied Earth Observation and Geoinformation, 45, 221–234, https://doi.org/10.1016/j.jag.2015.10.002, http://linkinghub.elsevier.com/retrieve/pii/S0303243415300386, 2016.

Malbéteau, Y., Merlin, O., Balsamo, G., Er-Raki, S., Khabba, S., Walker, J. P., Jarlan, L., Malbéteau, Y., Merlin, O., Balsamo, G., Er-Raki,
35 S., Khabba, S., Walker, J. P., and Jarlan, L.: Toward a Surface Soil Moisture Product at High Spatiotemporal Resolution: Temporally Interpolated, Spatially Disaggregated SMOS Data, Journal of Hydrometeorology, 19, 183–200, https://doi.org/10.1175/JHM-D-16-0280.1, http://journals.ametsoc.org/doi/10.1175/JHM-D-16-0280.1, 2018.

Merlin, O., Walker, J. P., Chehbouni, A., and Kerr, Y.: Towards deterministic downscaling of SMOS soil moisture using MODIS derived soil evaporative efficiency, Remote Sensing of Environment, 112, 3935–3946, https://doi.org/10.1016/j.rse.2008.06.012, 2008.

40 Merlin, O., Rüdiger, C., Al Bitar, A., Richaume, P., Walker, J. P., and Kerr, Y. H.: Disaggregation of SMOS soil moisture in Southeastern Australia, IEEE Transactions on Geoscience and Remote Sensing, 50, 1556–1571, https://doi.org/10.1109/TGRS.2011.2175000, 2012.

Merlin, O., Escorihuela, M. J., Mayoral, M. A., Hagolle, O., Al Bitar, A., and Kerr, Y.: Self-calibrated evaporation-based disaggregation of SMOS soil moisture: An evaluation study at 3km and 100m resolution in Catalunya, Spain, Remote Sensing of Environment, 130, 25–38, https://doi.org/10.1016/j.rse.2012.11.008, 2013.

45 Merlin, O., Malbéteau, Y., Notfi, Y., Bacon, S., Er-Raki, S., Khabba, S., and Jarlan, L.: Performance metrics for soil moisture downscaling methods: Application to DISPATCH data in central Morocco, Remote Sensing, 7, 3783–3807, https://doi.org/10.3390/rs70403783, 2015.

Molero, B., Merlin, O., Malbéteau, Y., Al Bitar, A., Cabot, F., Stefan, V., Kerr, Y., Bacon, S., Cosh, M., Bindlish, R., and Jackson, T.: SMOS disaggregated soil moisture product at 1 km resolution: Processor overview and first validation results, Remote Sensing of Environment, 180, 361–376, https://doi.org/10.1016/J.RSE.2016.02.045, https://www.sciencedirect.com/science/article/pii/S0034425716300736, 2016.

50 Moran, M. S., Clarke, T. R., Inoue, Y., and Vidal, A.: Estimating Crop Water Deficit Using the Relation between Surface-Air Temperature and Spectral Vegetation Index, Remote Sensing of Environment, 49, 246–263, https://naldc.nal.usda.gov/download/146/PDF, 1994.

Ojha, N., Merlin, O., Molero, B., Suere, C., Olivera-Guerra, L., Ait Hssaine, B., Amazirh, A., Al Bitar, A., Escorihuela, M., and Er-Raki, S.: Stepwise Disaggregation of SMAP Soil Moisture at 100 m Resolution Using Landsat-7/8 Data and a Varying Intermediate Resolution, Remote Sensing, 11, 1863, https://doi.org/10.3390/rs11161863, https://www.mdpi.com/2072-4292/11/16/1863, 2019.

Peng, J., Loew, A., Merlin, O., and Verhoest, N. E.: A review of spatial downscaling of satellite remotely sensed soil moisture, Reviews of Geophysics, 55, https://doi.org/10.1002/2016RG000543, 2017.

Rafi, Z., Merlin, O., Le Dantec, V., Khabba, S., Mordelet, P., Er-Raki, S., Amazirh, A., Olivera-Guerra, L., Ait Hssaine, B., Simonneaux, V., Ezzahar, J., and Ferrer, F.: Partitioning evapotranspiration of a drip-irrigated wheat crop: Intercomparing eddy covariance-, sap flow-, lysimeter- and FAO-based methods, Agricultural and Forest Meteorology, 265, 310–326, https://doi.org/10.1016/J.AGRFORMET.2018.11.031, https://www.sciencedirect.com/science/article/pii/S0168192318303848, 2019.

Sabaghy, S., Walker, J. P., Renzullo, L. J., Akbar, R., Chan, S., Chaubell, J., Das, N., Dunbar, R. S., Entekhabi, D., Gevaert, A., Jackson, T. J., Loew, A., Merlin, O., Moghaddam, M., Peng, J., Peng, J., Piepmeier, J., Rüdiger, C., Stefan, V., Wu, X., Ye, N., and Yueh, S.: Comprehensive analysis of alternative downscaled soil moisture products, Remote Sensing of Environment, 239, https://doi.org/10.1016/j.rse.2019.111586, 2020.

Sauer, T., Norman, J., Tanner, C., and Wilson, T.: Measurement of heat and vapor transfer coefficients at the soil surface beneath a maize canopy using source plates, Agricultural and Forest Meteorology, 75, 161–189, https://doi.org/10.1016/0168-1923(94)02209-3, https://www.sciencedirect.com/science/article/pii/0168192394022093, 1995.

Sellers, P. J., Heiser, M. D., and Hall, F. G.: Relations between surface conductance and spectral vegetation indexes at intermediate (100m2 to 15km2) length scales, J. Geophysical Research-atmospheres, 97, 19 033–19 059, 1992.